# Ultraflat Cu(111) foils by surface acoustic wave-assisted annealing

Bo Tian [1,5] ✉, Junzhu Li [1,2,5], Qingxiao Wang[3], Abdus Samad [2], Yue Yuan [2], Mohamed Nejib Hedhili[3], Arun Jangir[2], Marco Gruenewald [4], Mario Lanza [2], Udo Schwingenschlögl [2], Torsten Fritz [4], Xixiang Zhang [2] ✉ & Zheng Liu [1] ✉

Ultraflat metal foils are essential for semiconductor nanoelectronics applications and nanomaterial epitaxial growth. Numerous efforts have been devoted to metal surface engineering studies in the past decades. However, various challenges persist, including size limitations, polishing non-uniformities, and undesired contaminants. Thus, further exploration of advanced metal surface treatment techniques is essential. Here, we report a physical strategy that utilizes surface acoustic wave assisted annealing to flatten metal foils by eliminating the surface steps, eventually transforming commercial rough metal foils into ultraflat substrates. Large-area, high-quality, smooth 2D materials, including graphene and hexagonal boron nitride (hBN), were successfully grown on the resulting flat metal substrates. Further investigation into the oxidation of 2D-material-coated metal foils, both rough and flat, revealed that the hBN-coated flat metal foil exhibits enhanced anti-corrosion properties. Molecular dynamics simulations and density functional theory validate our experimental observations.

Metal foils with single crystallinity and ultraflat surface are considered ideal substrates for epitaxial growth of high-quality two-dimensional (2D) materials owing to their exceptional catalytic effects, lattice matching, and affordability[1,2]. In recent years, numerous studies in substrate engineering have been reported to enhance the quality of metal foils[3–5]. Previous work has achieved the single-crystallization of metal substrates through various methods, such as contact-free annealing[6], seeded growth of high-index-facet metals[7], and single-crystal sapphire epitaxy growth[8]. Using these single-crystal foils, large-area single-crystal hexagonal boron nitride (hBN), graphene, and 2D van der Waals heterostructures have been synthesized on Cu(110), Cu(111), Ni(111), and CuNi(111)[9–13], highlighting the substantial importance of metal substrate properties in producing high-quality nanomaterials. In addition to crystallinity, the surface flatness of metal foils is another crucial aspect that must be taken into consideration. The flatness of the substrate plays a vital role not only in high-temperature growth processes but also in the subsequent transfer of 2D materials. Commercial Cu foils typically possess rough surfaces, leading to various issues in the epitaxial growth of 2D materials, including high nucleation densities, excessive wrinkles, and film discontinuities[14–18]. While high-quality, ultraflat Cu(111) and CuNi(111) film wafers have been successfully produced through sputtering on sapphire substrates or chemical mechanical polishing processes[19,20], developing efficient flattening methods for metal foils remains crucial for achieving cost-effective production and scalable preparation of 2D materials. Currently, the electrochemical polishing (ECP) technique has been regarded as a widely accepted method to reduce metal-foil surface roughness through chemical reactions in the electrolysis process[21,22].

[1]School of Materials Science and Engineering, Nanyang Technological University, Singapore 639798, Singapore. [2]Physical Science and Engineering Division, King Abdullah University of Science and Technology (KAUST), Thuwal 23955-6900, Saudi Arabia. [3]Imaging and Characterization Core Lab, King Abdullah University of Science and Technology (KAUST), Thuwal 23955-6900, Saudi Arabia. [4]Institute of Solid State Physics (IFK), Friedrich Schiller University Jena, Jena 07743, Germany. [5]These authors contributed equally: Bo Tian, Junzhu Li. ✉e-mail: botianlab@gmail.com; xixiang.zhang@kaust.edu.sa; z.liu@ntu.edu.sg

However, we found that the ECP process introduces undesired contaminants into the shallow layer of the treated metal foils, resulting in additional impurities in the grown 2D materials, which is detrimental to their nanodevice applications[23–25]. Therefore, a contamination-free strategy for fabricating ultraflat metal foils is highly required.

In this study, we report a physical approach to flattening commercially available Cu foil surfaces using a surface acoustic wave (SAW) assisted annealing strategy. We found that the external SAW effectively facilitates the relocalization of the surface atoms of Cu foils under the annealing temperature. This leads to a considerable elimination of surface steps, further resulting in the formation of flat surfaces of the Cu(111) foils. The surface morphology of the foils produced was comprehensively evaluated, revealing the atomic-level steps and a significantly reduced surface roughness. The relocalization of Cu surface atoms during the SAW flattening process was modeled using molecular dynamics (MD) simulations. Furthermore, by utilizing the SAW-produced flat Cu foils as growth substrates, we synthesized smooth and continuous 2D-material films, including hBN and graphene. We also investigated the enhanced protective capabilities of hBN coatings against oxidation on flat metal surfaces, supported by experimental evidence and density functional theory (DFT).

## Results

### SAW flattening strategy

Commercial Cu foils typically exhibit rough surfaces, originating from the traditional cold-rolling process in metal foil production[26,27]. Annealing treatment is commonly used to flatten Cu foils and decrease their surface roughness to some extent[28]. Nevertheless, even after traditional annealing, the Cu foil still shows significant surface steps, failing to meet the criteria for an ideal substrate for the epitaxial growth of high-quality 2D materials. In our experiments, a SAW strategy was designed and implemented to eliminate the large steps and diminish the surface roughness of commercial Cu foils (Fig. 1a). The Cu foils were suspended on a custom-designed quartz shelf connected to a SAW generator, which was installed inside the quartz tube of the CVD system (Fig. 1b). The foils were then loaded into the high-temperature zone and heated to the desired annealing temperature. The SAW generator was activated, transmitting acoustic waves to the samples through a quartz rod under high-temperature conditions. After SAW-annealing treatment, the surface of the Cu foils became significantly smoother compared to both the as-received and traditionally annealed Cu foils (Fig. 1c–e). Optimizing the SAW treatment parameters was crucial for achieving a smooth Cu surface. To determine the optimal conditions, a series of controlled experiments were conducted, focusing on treatment duration, SAW power, and annealing temperature (Fig. 1f and Supplementary Fig. 1). In our study, we employed specific conditions with an annealing temperature of 1323 K, a treatment duration of 30 minutes, and a SAW power of 500 W. These parameters resulted in optimal surface smoothness of the Cu foil while minimizing excessive surface evaporation.

At the annealing temperature of 1323 K, slightly below the Cu bulk melting point (~1356 K)[29] but higher than the Cu(111) surface melting point (~1288 K)[30], the Cu foil behaves extremely softly and exhibits a quasi-liquid phase of the surface[31–34]. Therefore, the application of

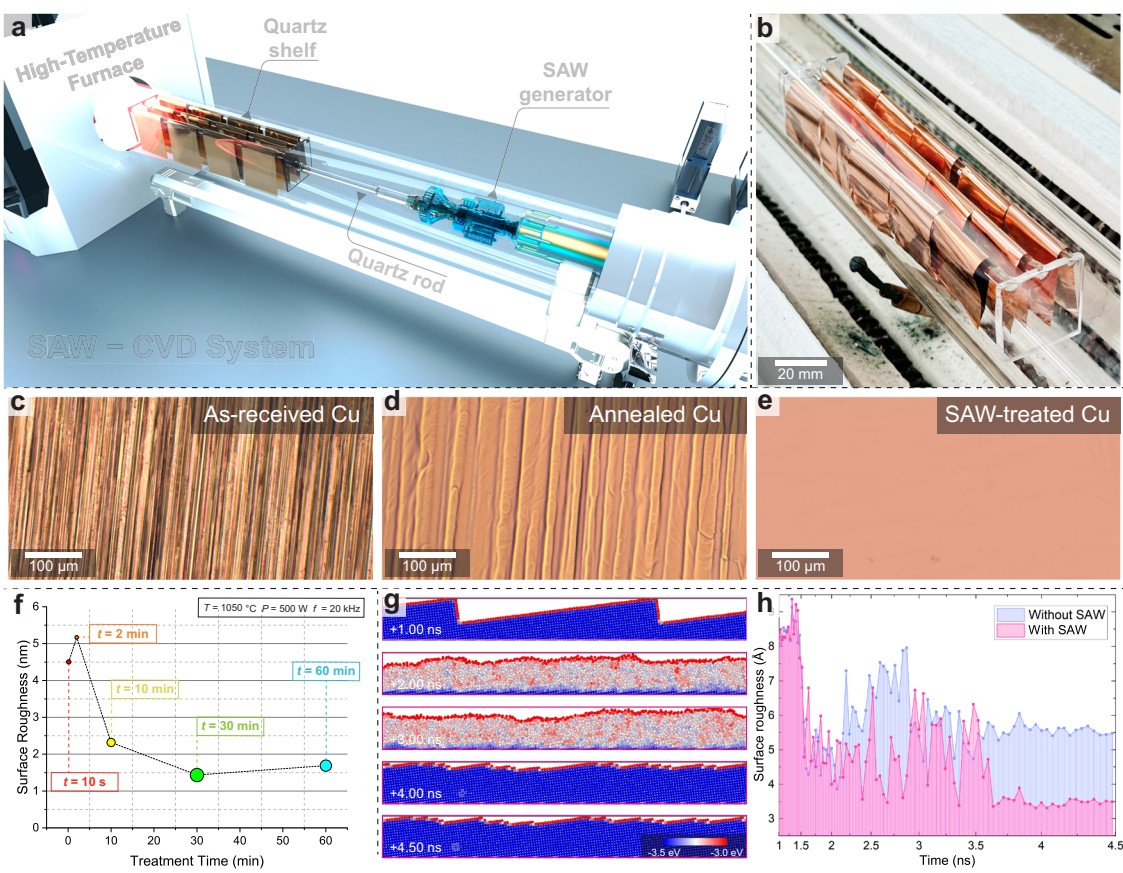

**Fig. 1 | Surface acoustic wave (SAW)-annealing treatments. a** Schematic of the SAW-annealing treatment system, consisting of the traditional chemical vapor deposition (CVD) system, the SAW generator, and the connected quartz shelf. **b** Photograph of treated Cu foils on the quartz shelf. **c–e** Large-area optical images of Cu surfaces captured from the as-received (**c**), the traditionally annealed (**d**), and the SAW-treated (**e**) Cu foils. **f** Surface roughness measured on produced Cu foils under different treatment durations during the SAW-annealing process. The annealing temperature, SAW power, and frequency are abbreviated as $T$, $P$, and $f$, respectively. **g** Molecular dynamics (MD) simulations of the Cu surface during SAW-annealing at 1323 K. The average potential energy of Cu atoms was calculated and color-coded. **h** Comparison of the surface roughness during traditional annealing (blue) and SAW annealing (red). The horizontal axis (time) is displayed on a nonlinear scale.

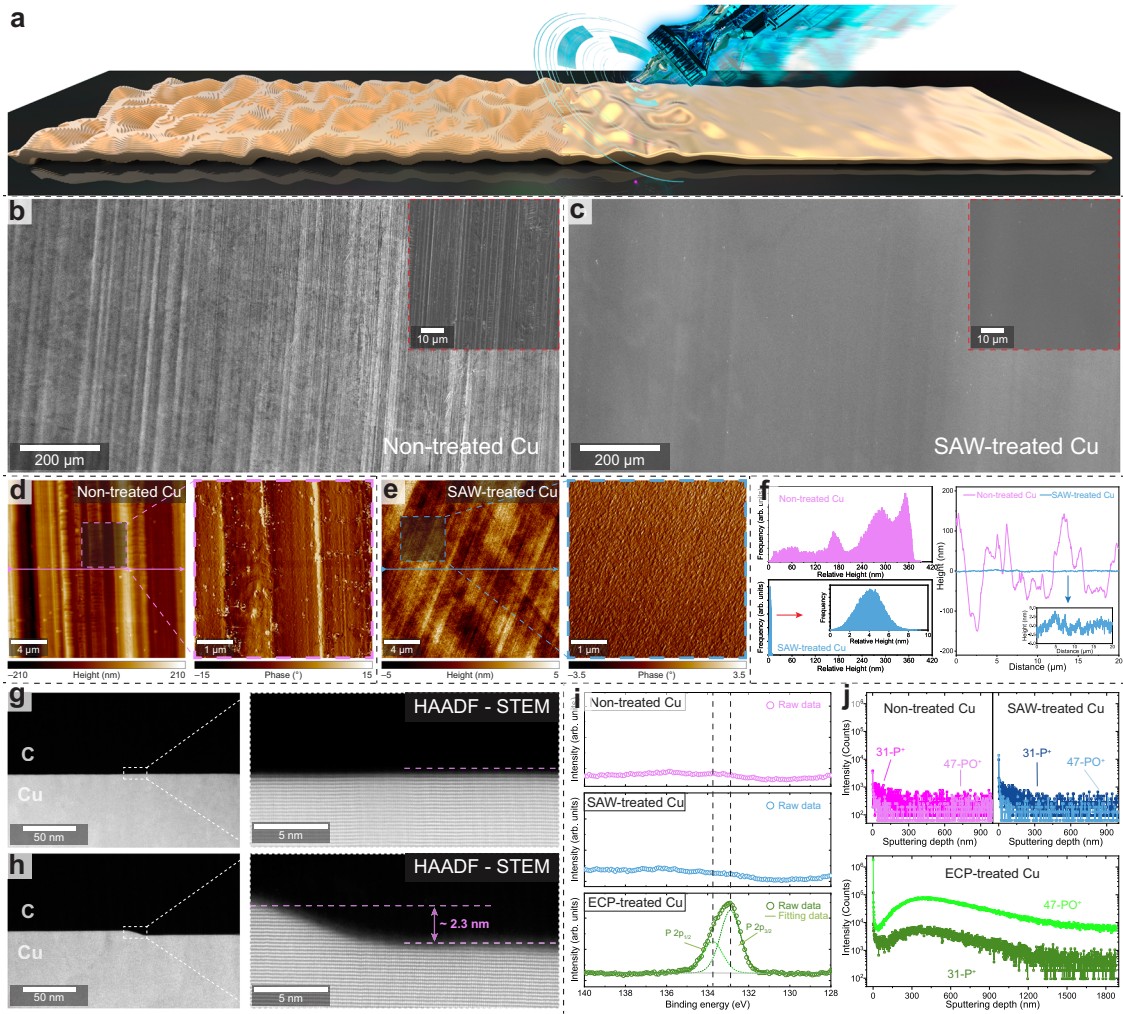

**Fig. 2 | Formation of flat Cu(111) foil via SAW treatments. a** Schematic of the SAW flattening process of the Cu foil surface. **b**, **c** Large-area scanning electron microscope (SEM) images of as-received non-treated Cu foil with the rough surface (**b**) and the SAW-treated Cu foil with the flat surface (**c**). **d**, **e** Atomic force microscopy (AFM) topography maps of non-treated (**d**, surface roughness of 53.1 nm) and SAW-treated Cu foil (**e**, surface roughness of 1.1 nm) in the measurement area of $20 \times 20\,\mu m^2$. The phase maps corresponding to the marked regions of $5 \times 5\,\mu m^2$ in the height maps are shown in the right panel. **f** Height distributions and line profiles for non-treated (pink) and SAW-treated (blue) Cu foils extracted from AFM maps in

**d** and **e**. **g**, **h** Cross-sectional high-angle annular dark-field scanning transmission electron microscopy (HAADF-STEM) images of the surface region of the produced flat Cu(111) foil, acquired from the flat region (**g**) and the step region (**h**). **i** X-ray photoelectron spectroscopy (XPS) spectra of P 2p core level collected from the non-treated (pink), SAW-treated (blue), and electrochemical polishing (ECP)-treated (green) Cu foils. The dashed lines indicate the peak positions of P $2p_{1/2}$ and $2p_{3/2}$ in the ECP-treated Cu. **j** Dynamic secondary ion mass spectrometry (D-SIMS) spectra of the P contaminant intensity in three types of Cu foils.

external SAW is proposed to facilitate the release of Cu atoms that are initially in higher energy states, thereby allowing these atoms to relocalize into more energetically favorable positions. This relocalization process reduces the surface step heights and leads to the formation of a flat Cu foil. To further understand the underlying mechanism by which the SAWs reduce the surface roughness during annealing, we performed MD simulations that tracked the energy states and movement of Cu atoms under the traditional and SAW-assisted annealing (Fig. 1g and Supplementary Fig. 2). The energy distribution revealed that most Cu atoms, except for those on the surface, exhibited low potential energy, while surface atoms had high potential energy due to reduced coordination and increased instability. As the system was heated, the crystal lattice began to degrade, leading to an increase in the potential energy of more atoms, as evidenced by the emergence of atoms with intermediate energies. The introduction of external vibrations through SAW treatment further enhanced this reconfiguration process. These vibrations induced additional energy fluctuations, particularly among surface atoms, facilitating their movement from unstable, high-energy sites to more stable, low-energy

sites. This increased atomic mobility led to a more significant reduction in surface roughness, producing a smoother and lower-energy surface compared to traditional annealing alone (Fig. 1h). Although MD simulations are performed on much shorter time scales than real-world processes due to computational limitations, they effectively reveal the atomic mechanisms through which SAW plays a crucial role in facilitating the flattening of the Cu surface. These results indicate that SAWs are an effective tool for manipulating atomic energy states and surface morphology, providing a controlled and efficient method for achieving smooth Cu foils.

## Flat Cu(111) foil via SAW annealing

A flat Cu(111) foil was produced using our proposed SAW strategy with optimized parameters (see Methods for experimental details, Fig. 2a). Large-area scanning electron microscope (SEM) images reveal the flat surface of the SAW-treated Cu foils, contrasting with the noticeably rougher surface of the commercial non-treated Cu foils (Fig. 2b–c). The uniform large-area electron backscatter diffraction (EBSD) inverse pole figure (IPF) color maps in normal and transverse directions

confirm the single-crystal nature of the produced flat Cu(111) foil (Supplementary Fig. 3). Atomic force microscopy (AFM) topography and phase maps show the rough surface of the non-treated Cu foil, exhibiting a surface roughness of 53.1 nm in the measurement area of $20 \times 20 \, \mu m^2$ (Fig. 2d and Supplementary Figs. 4-5). Following the SAW treatment, a remarkable reduction in surface roughness to 1.1 nm (more than 50-fold) is evident in Fig. 2e. The resulting Cu foil exhibited a uniformly smooth surface over a large area (Supplementary Fig. 6). Height distributions and line profiles extracted from AFM maps demonstrate the uniform surface and substantially reduced roughness of the SAW-treated Cu foil compared to the non-treated foil, as shown in Fig. 2f. It has been reported that normal annealing and single-crystallization processes can also improve substrate flatness[35]. Therefore, we compared the surface morphology of SAW-annealed Cu(111) foil with that of traditionally annealed foil and observed a significant reduction in surface roughness in the SAW-treated sample (Supplementary Fig. 7). Moreover, we prepared cross-sectional lamellae using focused ion beam (FIB) to further examine the surface of the SAW-treated Cu(111) foils. High-angle annular dark-field scanning transmission electron microscopy (HAADF-STEM) images reveal a generally flat surface on the produced Cu(111) foil (Fig. 2g). However, we observed that the surface is not perfectly atomically flat everywhere; in certain regions, steps with heights equivalent to several atoms are still present, as depicted in Fig. 2h.

It is worth noting that the flatness of the SAW-treated Cu foil is comparable to that of Cu foils treated using the ECP approach (refer to Methods for experimental details)[36,37]. However, considering the involvement of additional chemicals (e.g., urea and phosphoric acid) in the electrolyte of the traditional ECP process, we conducted further examinations to investigate potential contaminations in the ECP-treated Cu foils. High-resolution X-ray photoelectron spectroscopy (XPS) spectra of the phosphorus (P) 2p core level were collected from three samples: (i) as-received non-treated Cu foil, (ii) SAW-treated flat Cu foil, and (iii) ECP-treated Cu foil. It has been reported that post-annealing can remove some chemical contamination from ECP-treated Cu foil[37]. Therefore, to ensure a fair comparison in our study, the ECP-treated Cu foils were annealed prior to the XPS measurements (see Methods). An evident P 2p peak was detected from the ECP-treated Cu foil, which was absent in the spectra of the other two samples (Fig. 2i). To further examine the presence of P contaminants inside the Cu foil, dynamic secondary ion mass spectrometry (D-SIMS) and depth profiling experiments were conducted on three types of Cu foils. In alignment with the XPS results, a notable amount of P contaminants was observed in the subsurface of the ECP-treated Cu foil. In contrast, no additional increase of P content was observed at any depth in the SAW-treated Cu foil compared to the non-treated foil (Fig. 2j). Additionally, we investigated the presence of other potential contaminants, such as sulfur and nitrogen, in all Cu foils. XPS and D-SIMS measurements did not reveal any increase in the content of these elements following the SAW treatment, as shown in Supplementary Figs. 8 and 9. These results demonstrate that the ECP process introduces undesired chemical contaminants into the treated Cu foils, highlighting the importance of developing a contamination-free physically flattening technique.

## 2D material growth on flat Cu(111)
The smooth surface of the Cu foil is essential for the chemical vapor deposition (CVD) growth of large-area high-quality 2D materials. We synthesized wafer-scale monolayer graphene film on SAW-produced flat Cu(111) surfaces (Fig. 3a and Supplementary Fig. 10). After transferring on commonly used 4-inch 300-nm $SiO_2$/Si substrates, Raman spectra were collected from 400 predefined positions arranged in the 20 by 20 point array on the $5.5 \, cm \times 5.5 \, cm$ graphene film (Fig. 3b). These spectra exhibit the characteristic G and 2D bands of graphene, with the absence of the evident D band, indicating the high quality of

as-grown monolayer graphene film. Statistical analysis of the 2D FWHM and $I_D/I_G$ ratios confirms the adlayer-free nature and exceptional quality of the monolayer graphene across the entire film area (Fig. 3c). Raman maps of the $I_D/I_G$ ratio and 2D FWHM, acquired from nine randomly selected locations across the graphene film, demonstrate a uniform monolayer of high quality, free from noticeable adlayers and defects (Supplementary Fig. 11). These characterizations are consistent with previous observations of CVD-grown, high-quality, adlayer-free graphene films[2,19,38,39].

Additionally, to compare the as-grown graphene on Cu foils with varying surface roughness, we synthesized single-crystal graphene islands on both rough and flat Cu(111) surfaces using the CVD approach (see Methods for experimental details). Subsequently, both samples were exposed to natural oxidation in air for a duration of seven days (Fig. 3d). The SEM image, obtained from graphene islands grown on the rough Cu surface, depicts folds and cracks in the graphene, along with distinct surface steps on the Cu substrate (Fig. 3e). Dark regions were observed near graphene folds, indicating the Cu oxidation occurrence beneath the graphene island. In contrast, the graphene island grown on a flat Cu(111) foil displays a highly smooth film surface devoid of noticeable folds and cracks, as shown in Fig. 3f. Yet, we noticed small darker areas at the edges of the island, as depicted in the enlarged SEM image, indicating Cu oxidation still occurred in these graphene-covered regions. Consequently, we further examined the oxidation in the same area by conducting Raman measurements. Uniform Raman maps of the intensity of G and 2D bands, as well as the full width at half maximum (FWHM) of the 2D band, reveal the high crystal quality, absence of folds, and adlayer-free nature of the as-grown graphene on the flat Cu(111) foil (Supplementary Fig. 12). The Raman map of the $Cu_xO$ band intensity illustrates complete oxidation of the Cu surface in uncovered regions (Fig. 3g). True component analysis of the large-area Raman mapping provides clear visualization of $Cu_xO$ presence, particularly at the island edges, as confirmed by the corresponding Raman spectra (Fig. 3h-i). We suggest that this oxidation at the edges of graphene islands is induced by the galvanic corrosion mechanism in the graphene-coated Cu system[40-42]. It has been reported that bilayer graphene can solve the electrochemical corrosion problem and enhance anti-corrosion properties by blocking water and oxygen from reaching the Cu surface[43]. In our study, to avoid the influence of this accelerated corrosion effect, we synthesized an insulating and inert 2D material, i.e., hBN, on both rough and flat Cu(111) surfaces (Fig. 3j). Distinct Cu steps were observed in the optical image of hBN islands grown on a rough Cu foil. In contrast, the hBN island on a flat Cu(111) foil exhibited a uniform and relatively smooth surface.

## Anti-corrosion properties of hBN-coated Cu(111)
To further study the oxidation of Cu foil under the protection of a 'complete' 2D-material film, we synthesized monolayer hBN on commonly used rough and SAW-treated flat Cu(111) substrates. All samples underwent the water-assisted oxidation experiment for 48 hours (see Methods for experimental details, Fig. 4a and Supplementary Fig. 13). Photographs and optical images revealed a noticeable color difference between the hBN/R-Cu and hBN/F-Cu samples after the oxidation treatment (Fig. 4b, c). A typical Raman spectrum collected from the hBN/R-Cu surface exhibits distinct bands at 148, 216, 299, 413, 500, 649, and 798 $cm^{-1}$, respectively (Fig. 4d), consistent with reported Raman bands of $Cu_2O$[36,44]. In contrast, no $Cu_xO$ bands were observed in a typical Raman spectrum collected from the hBN/F-Cu sample. High-resolution XPS spectra of Cu 2p core levels were further acquired from the hBN/R-Cu surface, revealing broader Cu $2p_{3/2}$ and Cu $2p_{1/2}$ peaks (Fig. 4e), corresponding to peaks of $Cu_2O$ at 932.4 and 952.2 eV, and $Cu(OH)_2$ at 934.5 and 954.3 eV[45]. Sharp Cu 2p peaks were detected from the hBN/F-Cu sample, with binding energies of 932.6 and 952.4 eV (Fig. 4f), corresponding to Cu $2p_{3/2}$ and Cu $2p_{1/2}$[45,46].

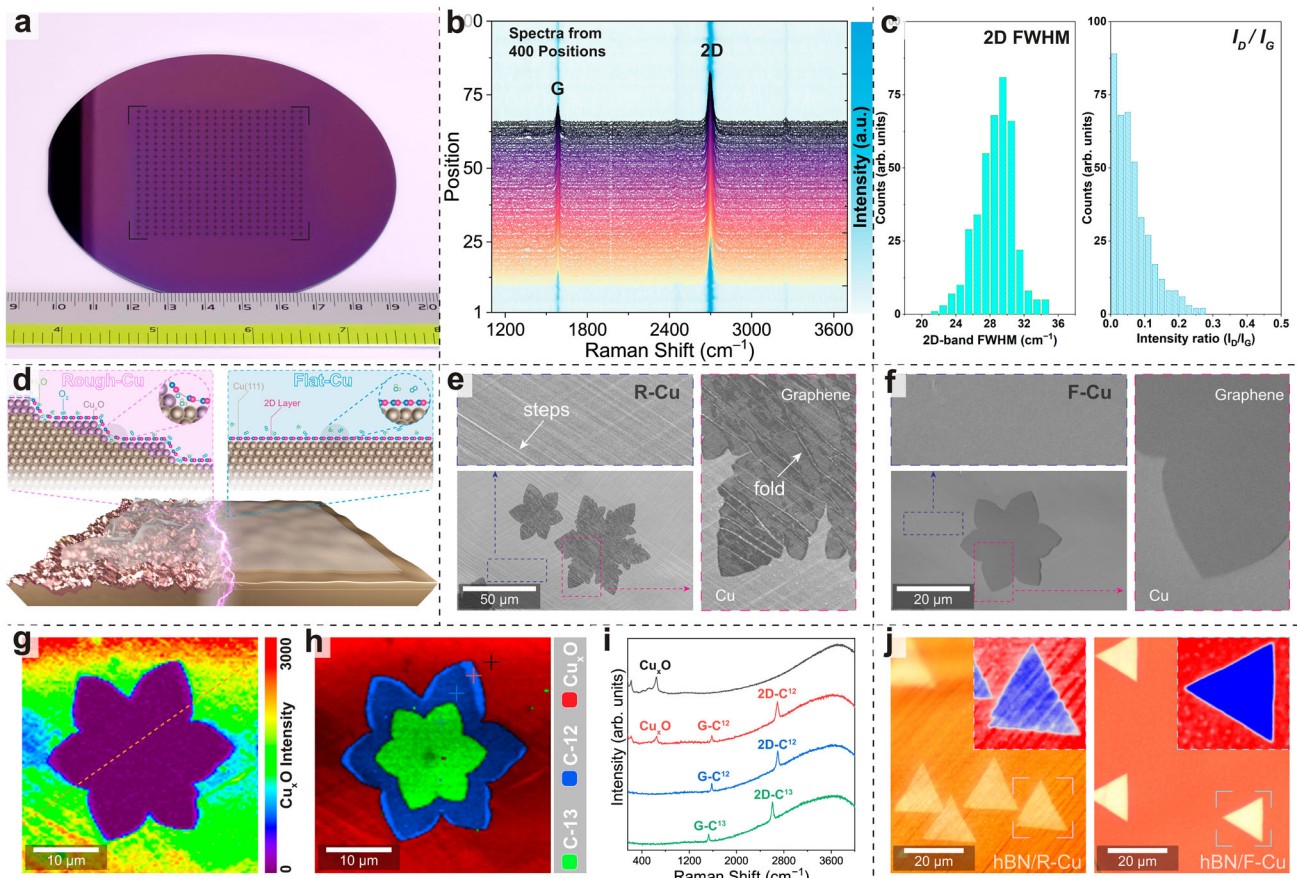

**Fig. 3 | 2D-material growth on SAW-treated flat Cu foils. a** Photograph of graphene film grown on SAW-treated flat Cu(111) foil, subsequently transferred to a 4-inch SiO₂/Si wafer. **b** Raman spectra collected from 400 positions (20 × 20 array) on the graphene film shown in (**a**). **c** Distribution of 2D FWHM and $I_D/I_G$ ratios from the monolayer graphene film, collected at 400 positions across the entire sample. **d** Schematic of wrinkled and smooth 2D-material films grown on Cu foils with rough (R-Cu, left) and flat (F-Cu, right) surfaces. **e**, **f** SEM images of as-grown graphene islands on commonly used rough Cu foil (**e**) and SAW-treated flat Cu foils (**f**). Cu steps and graphene folds are indicated by arrows. **g** Raman intensity map of Cu$_x$O bands. An intensity line scan along the dashed line is provided in Supplementary Fig. 12. **h** True component Raman map of as-grown graphene island. **i** Raman spectra collected from four marked positions in (**h**). **j** Optical images of as-grown single-crystal hBN islands on rough (left) and flat (right) Cu foils. The insets show magnified false-color images highlighting the marked hBN islands.

Additionally, we conducted an analysis of the Cu LMM Auger spectra on both hBN/R-Cu and hBN/F-Cu samples (Fig. 4g). A distinct peak at 916.79 eV kinetic energy was observed in the hBN/R-Cu sample, suggesting the predominance of the +1 oxidation state for Cu. Whereas in the hBN/F-Cu sample, a sharp peak at 918.61 eV kinetic energy was detected, confirming the Cu in the metallic state[45]. For further study on the oxidation status, we acquired scanning tunneling microscope (STM) images on the hBN-covered Cu surface around the step-edge region (Fig. 4h). A perfect honeycomb structure of the hBN lattice was observed on the flat Cu surface in the atomically resolved low-temperature STM (LT-STM) image acquired at 4.5 K. Compared to the flat region, the atomic lattice of hBN is much more pronounced near the edge region, suggesting the decoupling of the hBN film from the Cu surface at the step edge. These results suggest that the continuous monolayer hBN film grown on flat Cu(111) foil can effectively protect the Cu surface from oxidation. Furthermore, DFT calculations were employed to investigate the oxidation mechanism of the Cu(111) surface by determining the adsorption energy ($E_{ads}$) of an H₂O molecule (Fig. 4i). The calculated $E_{ads}$ on a bare flat Cu surface was found to be −0.53 eV. As shown in Fig. 4j, it increased to 0.73 eV when the flat Cu surface was covered by monolayer hBN, because the gap between flat Cu and hBN is too small to accommodate the H₂O molecule and hBN thus has to form a bulge. This suggests that H₂O molecules cannot diffuse into the gap between flat Cu and hBN, implying that the hBN-covered flat Cu surface is less prone to be oxidized than the bare flat Cu surface. In other words,

corrosion protection is provided by monolayer hBN, aligning with our experimental findings. This superior anti-corrosion system, i.e., hBN on flat Cu(111), is expected to serve as an optimal epitaxial substrate for the growth of other high-quality single-crystal 2D materials, such as graphyne, black phosphorus, borene, and metallene.

## Discussion

A physical flattening strategy is reported for producing flat metal foils. We found that SAW accelerates the relocalization of metal surface atoms, thereby effectively eliminating larger steps and ultimately achieving contamination-free flattening of the foils. Elaborating on the flat Cu(111) foil, we fabricated large-area, high-quality 2D materials, including graphene and hBN. Additionally, we observed that the flat Cu(111) foil yielded by our proposed SAW strategy, covered with a continuous hBN layer, exhibited superior anti-corrosion properties, making it an ideal substrate for the future epitaxial growth of larger-area single-crystal 2D materials. The proposed chemical-contamination-free SAW flattening strategy is expected to be applicable to a wide range of metals, offering significant potential for applications in metal surface engineering and industrial production.

## Methods
### Fabrication of Cu(111) foil

The as-received Cu foil (Alfa Aesar) was pre-cleaned using an ammonium persulfate solution, followed by rinsing with deionized water and

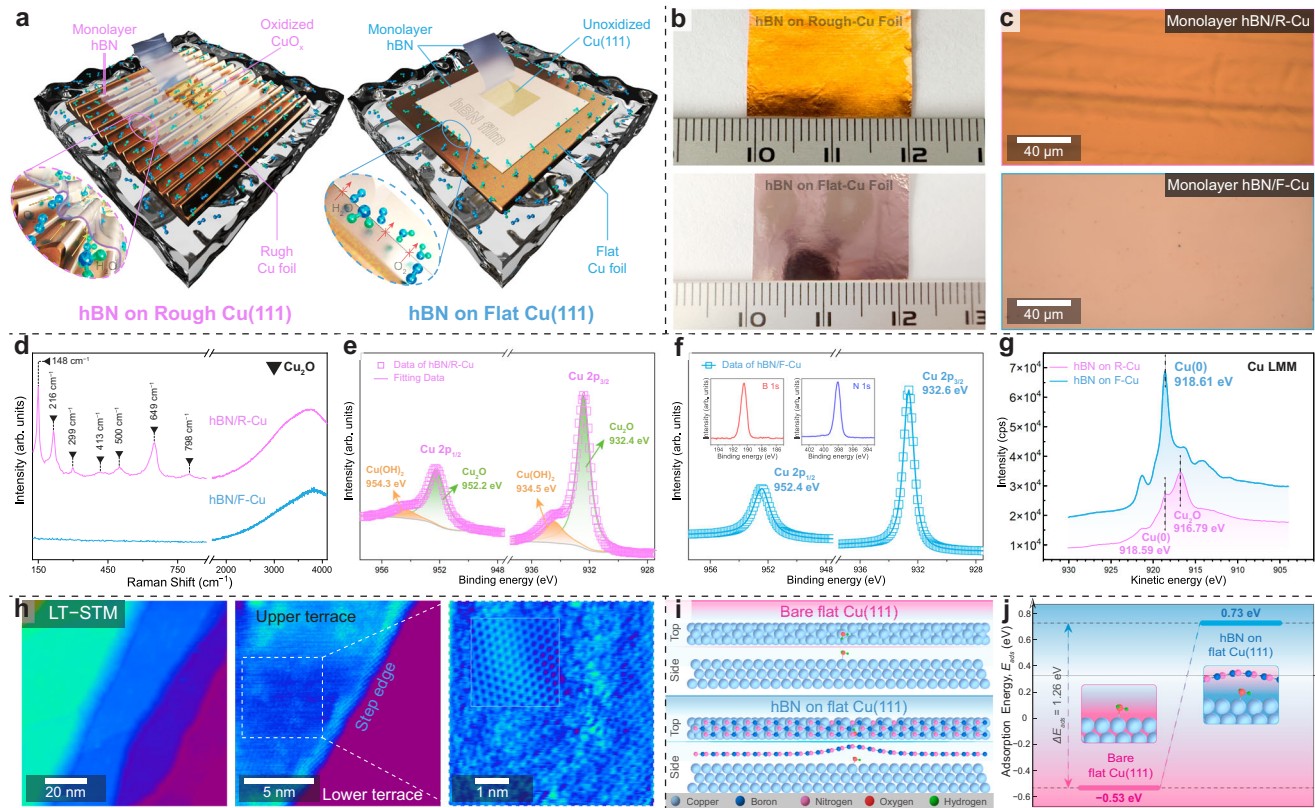

**Fig. 4 | Anti-corrosion properties of monolayer hBN on flat Cu(111) foil.**
**a** Schematic illustrations of water-assisted oxidation experiments for rough (left) and flat (right) Cu foils covered with monolayer hBN. **b, c** Photographs and optical images of hBN monolayers on rough (top) and flat (bottom) Cu foils after the oxidation experiment. **d** Typical Raman spectra of hBN on rough (magenta) and flat (blue) Cu foils. **e** XPS spectra of Cu 2p core levels collected from hBN-covered rough Cu foil. **f** XPS spectra of Cu 2p core levels collected from hBN on the flat Cu foil. The XPS spectra of B 1 s and N 1 s are shown in insets. **g** High-resolution Cu LMM Auger spectra from hBN/R-Cu (magenta) and hBN/F-Cu (blue). **h** Low-temperature scanning tunneling microscope (STM) images of hBN on Cu surface ($V = 0.1$ V, $I = 10$ pA, $T = 4.5$ K). The right panel shows the atomically resolved detailed image with a fast Fourier transform (FFT)-filtered inset in a dimension of $2.5 \times 2.5$ nm$^2$. **i** Top and side views of the atomic structures used for studying $H_2O$ adsorption on bare flat (top) and hBN-covered flat (bottom) Cu surfaces. The Cu, B, N, O, and H atoms are shown in cyan, blue, magenta, red, and green, respectively. **j** Adsorption energies of $H_2O$ on the different Cu surfaces.

isopropyl alcohol. This cleaning step is essential to remove surface contaminants and prepare the foil for subsequent processing. After cleaning, the Cu foil underwent a single-crystallization process via the contact-free annealing approach[6]. This involved a long-time annealing procedure under a controlled gas flow of $H_2$ (50 sccm) and Ar (350 sccm), which facilitated the transformation of the foil into a single-crystal Cu(111) foil.

### SAW treatment of Cu foil
The SAW treatment process was conducted using a custom-designed SAW-CVD system that integrates a SAW generator with an automated CVD growth system. This system produces acoustic waves at a frequency of 20 kHz, with adjustable power levels up to 700 W. The SAW generator is connected to a quartz boat or shelf via a quartz rod, with all components pre-fused together to ensure a stable and secure connection during high-temperature operations. The prepared Cu(111) foil was placed in the quartz boat or shelf, depending on the number of samples being treated. This assembly was then connected to the SAW generator. The sample was slid into the CVD system and heated to an annealing temperature ranging from 1000 °C to 1070 °C. This temperature range was carefully selected to induce a quasi-liquid phase on the Cu foil surface, promoting surface smoothing through SAW treatment. Once the target temperature was reached, the sample was maintained at this temperature for 30 minutes to achieve thermal equilibrium. Following this stabilization period, the SAW generator was activated to apply acoustic waves to the Cu foil while maintaining the high annealing temperature. The duration of the SAW treatment

varied from 10 seconds to 60 minutes, depending on the specific experimental conditions. The frequency of the SAWs was kept constant at 20 kHz, with power adjustable from 50 to 700 W. These parameters were optimized through controlled experiments to balance energy input with the desired surface modification effects. After the SAW treatment, the generator was shut down, and the sample was rapidly removed from the high-temperature zone, allowing it to cool naturally to room temperature.

### ECP treatment of Cu foil
We employed the conventional ECP method to treat metal surfaces. The as-produced single-crystal Cu(111) foils and a pre-cleaned commercial Cu foil were immersed in the electrolytic polishing solution, which comprised phosphoric acid (300 ml), ethanol (100 ml), isopropyl alcohol (20 ml), and urea (2 g). A constant voltage of 3 V was applied between the target single-crystal Cu(111) foils (anode) and standard Cu foils (cathodes) to initiate the electrolytic polishing process. Following electrolytic polishing, the Cu(111) foil was thoroughly rinsed with deionized water and isopropyl alcohol. Ultimately, a smooth surface of the single-crystal Cu(111) foil was obtained. For a fair comparison, we annealed the Cu foil in the CVD system under $H_2$ (50 sccm) and Ar (50 sccm) for 1 hour following the ECP treatment.

### CVD growth of graphene and hBN
A graphene CVD (G-CVD) growth system was utilized to synthesize graphene on Cu foils with various surface roughness. Initially, the G-CVD system was evacuated to approximately 0.1 Pa after loading the

Cu foil substrate, which had been prepared as a Cu package. Subsequently, the furnace was heated to 1030 °C, and the sample was automatically transferred from the room-temperature region to the high-temperature region and held for 5 minutes to stabilize the temperature. The $H_2$ gas flow was maintained at 10 sccm. Following temperature stabilization, $CH_4$ gas (5–10 sccm) was introduced to initiate graphene growth. Upon completion of the growth process, the samples were transferred from the high-temperature zone to the room-temperature zone to allow rapid cooling while maintaining the same gas flows. Eventually, the graphene films were grown on the Cu substrate.

For the isotopic labeling graphene growth experiments, commonly used $^{12}C$ methane and $^{13}C$ isotopic labeling methane gases were used as carbon sources in CVD growth under controlled conditions. During the graphene synthesis, 5 sccm of methane with either $^{13}C$ or $^{12}C$ was alternately introduced into the growth chamber, with each isotopic growth period lasting 10 minutes to allow sufficient time for graphene formation from each carbon source. This alternation between $^{13}C$ and $^{12}C$ was repeated multiple times to produce distinct isotopically labeled graphene regions. Throughout the process, the furnace temperature was kept at 1030 °C, and $H_2$ gas was supplied at a constant flow rate of 20 sccm. After the growth process, the sample was slid into the room-temperature zone.

A CVD growth system equipped with two tube furnaces was employed for the growth of single-crystal hBN islands and monolayer films. Ammonia borane ($H_3N\text{-}BH_3$, 95%, Sigma-Aldrich) served as the precursor for hBN growth. The temperature in the growth area of the furnace was raised to 1050 °C, while the precursor temperature was maintained at 85–95 °C. Subsequently, the Cu package substrate was inserted into the high-temperature region, initiating the hBN growth process by introducing $H_2$ (10 sccm) and Ar (5–15 sccm). Upon completion of the growth process, the samples were transferred from the growth zone to the room-temperature zone for rapid cooling. Finally, the hBN films were obtained on the Cu substrate.

## Water-assisted oxidation experiment

The water-assisted oxidation experiments were conducted on samples of as-grown hBN on rough and flat Cu foils. Initially, the fresh hBN/Cu samples were placed on the surface of deionized water in a glass container. The container was sealed, and the entire setup was then placed in an oven maintained at a temperature of 60 °C for a duration of 48 hours for the oxidation treatment.

## Optical, Raman, AFM, and SEM Characterizations

Optical and Raman measurements were performed using a confocal Raman spectroscopy system (Alpha 300 R, WITec) with laser wavelengths of 488 and 532 nm. The topography maps of the Cu foil surface were obtained using the AFM (Dimension Icon, Bruker). SEM images and EBSD maps were captured using an environmental SEM (Quattro ESEM, Thermo Fisher Scientific).

## D-SIMS, XPS, and STM Measurements

D-SIMS experiments (Hiden Analytical Company, UK) were performed under ultrahigh vacuum conditions, typically reaching a pressure of $10^{-9}$ Torr. A continuous $Ar^+$ ion beam with an energy of 4 keV was employed to sputter the sample, while the desired ions were sequentially collected using a MAXIM spectrometer equipped with a quadrupole analyzer. To mitigate edge effects, ions were extracted from a specific area (typically $200 \times 200 \, \mu m^2$ in size), positioned at the center of the sputtered region, estimated to be $750 \times 750 \, \mu m^2$, using precise electronic gating. The conversion of sputtering time to sputtering depth was conducted assuming a consistent sputtering rate, with adjustments made for crater depth measurements using the stylus profiler DektakXT from Bruker. XPS studies were carried out in a Kratos Axis Supra DLD spectrometer equipped with a monochromatic Al Kα X-ray source (hν = 1486.6 eV) operating at 75 W, a multi-channel plate and delay line detector under a vacuum of -$10^{-9}$ mbar. All spectra were recorded using an aperture slot of $300 \times 700 \, \mu m^2$. Survey spectra were collected using a pass energy of 160 eV and a step size of 1 eV. A pass energy of 20 eV and a step size of 0.1 eV were used for the high-resolution spectra. Low-temperature STM measurements were conducted using a JT-STM/AFM system from SPECS Surface Nano Analysis GmbH, operated with a Pt/Ir tip.

## FIB, TEM, and STEM Measurements

The cross-sectional lamellae were prepared using a FIB (Helios G4, Thermo Fisher Scientific). To prevent surface damage during the ion beam milling, a protection layer was first coated inside FIB using electron-beam-assisted carbon deposition, electron-beam-assisted Pt deposition, and ion-beam-assisted Pt deposition. The thickness of the prepared lamella was less than 50 nm. TEM and STEM were performed on a double aberration-corrected transmission electron microscope (Titan Themis Z, Thermo Fisher Scientific) operated at 300 kV.

## MD and DFT simulations

MD simulations were conducted employing the large-scale atomic/molecular massively parallel simulator[47]. The simulation system consisted of a Cu surface free along the z-direction within a box periodic along the x and y-directions. An embedded atom model force field was used. The timestep for the MD simulations was set to 1.0 fs. Initially, the system was equilibrated at 298 K for 1 ns using a Nosé-Hoover thermostat. Following this, the temperature was ramped from 298 K to 1323 K linearly over 0.5 ns. Upon reaching the peak temperature, the system was equilibrated for an additional 2 ns, both with and without external SAW. For the SAW simulation, the frequency and amplitude were set to 0.2 THz and 0.1 Å, respectively. Although in experiments the external vibration has a much lower frequency (-20 kHz) and a larger amplitude (-100 μm), these parameters were adjusted to account for the limitations in length and time scales inherent to MD simulations. The system was then cooled down to 298 K over 0.5 ns and allowed to relax at this temperature for additional 0.5 ns to achieve final surfaces. Throughout the simulation, the potential energy of each atom was averaged considering the value every 1000th step, and these averages were used to generate a color-coded atomic representation. The roughness was monitored throughout the simulation by calculating the average surface depth.

DFT calculations were carried out using the projector augmented wave method of the Vienna ab initio simulation package[48] with Perdew-Burke-Ernzerhof exchange-correlation functional[49]. Van der Waals corrections by the DFT-D3 method were employed. An energy cutoff of 500 eV was used for the plane wave expansion. A k-mesh of $1 \times 13 \times 1$ was employed for all the calculations. Flat Cu(111) surface models were constructed without and with hBN coverage adopting a supercell with lattice parameters $62.27 \times 2.57$ Å. An $H_2O$ molecule was added to each model. The models then were relaxed with a force criterion of −0.01 eV/Å, fixing all Cu atoms except for those at the surface at face-centered cubic bulk positions. The $H_2O$ adsorption energy was calculated as $E_{ads} = E_{Cu+H2O} - E_{Cu} - E_{H2O}$ without hBN coverage and as $E_{ads} = E_{hBN/Cu+H2O} - E_{hBN/Cu} - E_{H2O}$ with hBN coverage, where $E$ denotes the total energy.

## Data availability

Relevant data supporting the key findings of this study are available within the article and the Supplementary Information file. All raw data generated during the current study are available from the corresponding authors upon request.

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

## Acknowledgements

This project was supported by the National Research Foundation Singapore (NRF-CRP22-2019-0007, NRF-CRP26-2021-0004), DSO National Laboratories under the AI Singapore Program (AISG2-GC-2023-009),

and A*STAR under its MTC Programmatic Grant (M23M2b0056) (Z.L.). This work was also supported by the King Abdullah University of Science and Technology (KAUST), under the Semiconductor Initiative – Emerging Semiconductor Materials Thrust (X.Z.).

## Author contributions

B.T. conceived the experiments. B.T., X.Z., and Z.L. supervised the project. B.T. and J.L. performed the Cu annealing and 2D materials growth experiments. B.T., J.L., and M.N.H. performed the Raman, SEM, EBSD, D-SIMS, and XPS characterizations. Y.Y. and M.L. performed the AFM measurements. B.T. and Q.W. performed the FIB and STEM measurements. M.G. and T.F. performed the STM measurements. A.S., A.J., and U.S. performed the MD and DFT simulations. B.T. and J.L. wrote the manuscript. All coauthors discussed and provided comments on the manuscript.

## Competing interests

The authors declare no competing interests.
