## [Transparent Peer Review file · Nature Communications]

Ultraflat Cu(111) Foils by Surface Acoustic Wave Assisted Annealing

Corresponding Author: Professor Zheng Liu

Version 0:

Reviewer comments:

Reviewer #1

(Remarks to the Author)

This manuscript reports a surface acoustic wave (SAW) assisted annealing strategy to flatten Cu foil surfaces without polluting. Relying on the SAW-annealing technique, smooth and continuous graphene and hBN films are obtained. The flatness of the substrate plays crucial roles in not only high-temperature growth but also post-growth peeling-transferring of 2D materials. Although ultra-flat Cu(111) and CuNi(111) wafers have been proposed via sputtering on sapphire (Nano Lett. 2020, 20 (9), 6798-6806) or CMP process (Adv. Mater. 2015, 27 (8), 1376-1382), mirror-like metal foils have not been reported so far. The pollution-free treatment of transforming commercial rough metal foils into ultra-flat substrates is an important issue. The SAW-annealing technique seems to pave the way for CVD-compatible ultra-flat growth of 2D materials. However, the authors paid much attention to the oxidation behaviour of graphene and hBN-coated Cu foils, including natural oxidation (Fig. 2), water-assisted oxidation (Fig. 3), and interfacial oxidation (Fig. 4). The DFT calculation and corresponding theoretical analysis are also about oxidation mechanism (Fig. 5). These extensive characterizations and analyses on Cu oxidation have crowded out the descriptions of SAW-annealing technique and the ultra-flat growth of high-quality graphene and hBN. Experimental details and mechanisms of SAW-annealing are unclear, and the quality evaluations on the resultant graphene and hBN are very limited. The title is no longer applicable to the content of the manuscript. I strongly recommend the authors reorganize the manuscript and put more space for SAW-annealing and ultra-flat growth of 2D materials. A list of the urgent issues follows below:

1. The sentence "External SAW facilitates the release of Cu atoms in higher energy states, enabling them to relocate to more energetically favorable states" seems to explain the mechanism of the SAW strategy. However, no actual evidence has been presented, and the MD simulations in Supplementary Fig.2 only exhibited the result of SAW treatment. The process of the SAW in Fig.1a is not scientific enough. More details about SAW strategy should be proposed (e.g. energy states of Cu atoms?).
2. It is proverbial that normal annealing and single-crystallization processes can also improve the flatness of the substrate (ACS Nano 2022, 16 (1), 285-294.), the authors should give more tangible evidence about the superiority of SAW compared with traditional annealing treatment and detailed process so that peers can evaluate the reproducibility of the experiment.
3. Terms such as "high-quality 2D materials", "large-area", "high crystal quality", "absence of folds", and "adlayer-free" are present several times in this manuscript, but the available evidence is insufficient to prove it. The present characterization area is too small to show its advanced nature (in Fig.2b,c, Fig.2e,f). The authors should provide characterization results on a larger scale and compare with the result on single-crystal Cu(111) foils (Adv. Mater. 2019, e1903615), CuNi(111) foils (Nature 2021, 596 (7873), 519-524), Cu(111) wafers (Nano Lett. 2020, 20 (9), 6798-6806), and CuNi(111) wafers (Nature 2020, 577 (7789), 204-208).
4. More characterizations should be provided to prove the surface uniformity of the whole foils. At least several specified regions should be measured by AFM to give statistical results and to avoid accidental errors.
5. In the DFT calculations, the authors froze all of the Cu atoms. To give expression to possible surface reconstruction and surface tension, Cu atoms on the surface should be relaxed sufficiently. In addition, the cutoff energy of 400 eV needs to be tested to obtain reliable results for this system with atoms such as N, O, and so on.
6. The parameters of theoretical calculation should be clarified. For example, the SAW information of MD calculation needs to be described. Moreover, the supercell size and k-point mesh of DFT calculation are necessary to be mentioned.

Reviewer #2

(Remarks to the Author)

The author innovatively applied the SAW method to promote the movement of Cu atoms at high temperatures to achieve the flattening of Cu foils. As a demonstration of the application, the authors grew hBN on the ultraflat Cu foils and found that it had good anticorrosion properties. The author solves the long-standing problem of uneven substrates in large-scale 2D material production and avoids the introduction of additional impurities, which is of high significance. I have the following comments, hoping to make this article better comply with the publishing standards of Nature Communications:

1. The most important thing is that the innovation of this article is that it proposes a new SAW treating method, but the discussion of the SAW treating process itself is very short. How do the temperature during SAW treatment, the length of treating time, the frequency and power of SAW, etc. affect the treating effect? What are the optimized specific parameters? The authors should provide the corresponding experiments in detail, and the method section should also be supplemented.
2. The author compares the impurity content of Cu foil after ECP treatment with that of Cu foil after high-temperature annealing, which is unfair. To grow 2D materials, the treated Cu foil will be subjected to high temperature for CVD growth, and many impurities may volatilize or react during the high-temperature process. Will the Cu foil that is ECP treated first and then high-temperature annealed have a flatter surface and a lower or similar impurity content?
3. In the analysis in Fig. 5 and the schematic diagram in Fig. S1, the authors implied that the SAW-treated Cu foil has only 1-atom steps. More experiments are needed to prove this, including atomically resolved cross-sectional STEM images.
4. According to the literature, flat Cu foil can prevent corrosion itself (ref. 8). Can the authors compare the anticorrosion effects of rough Cu, flat Cu, and flat Cu with hBN?
5. According to the literature, bilayer graphene can solve its electrochemical corrosion problem (Nature Communications, 2023, 14(1): 7447). As the authors claimed that the oxidation at the edges of graphene islands is induced by galvanic corrosion, would bilayer graphene have a better performance?
6. Some minor problems, the font size of Fig. 2d is too small; Fig. 3a lacks the corresponding labels and illustrations of each part.

Reviewer #3

(Remarks to the Author)

The authors present results on the smoothing of catalytic copper foils during 2D materials synthesis in furnaces using a surface acoustic wave technique.

The authors demonstrate that the SAW treatment provides an additional annealing effect on the smoothness of the catalytic substrates, with supporting evidence from AFM and cross-sectional TEM, as well as Raman spectroscopic investigation of isotopically differentiated growths and EBSD crystal orientation investigations.

The authors also demonstrate that while the SAW results are comparable to the more widely employed electrochemical polishing techniques, there is no unwanted contamination introduced thanks to the lack of an electrolyte. They demonstrate this with XPS and D-SIMS spectroscopies.

Finally, the authors use the 2D materials layers in anticorrosion studies to determine the rate of diffusion of oxygen under flakes, used as a proxy for the strength of the surface binding, and supporting the work with Raman mapping and HAADF-STEM, HRTEM and EELS.

The experimental work is supported by DFT simulations of water binding to the Cu(111) surface, coated with hBN, coated with graphene over steps and bare.

Overall, the work is detailed, interesting and the conclusions are very well supported by the comprehensive experimental and simulation data. The technique presented shows strong benefits for the production of 2D materials on metal catalyst layers by CVD. I would be delighted to recommend publication after the following issues are addressed:

1. Please provide much more method detail on the SAW device, coupling to the system, commercial vs. homebrew, power, time, etc. Details are almost completely absent from the manuscript, and the study as it stands would be impossible to replicate or verify.
2. The isotopically differentiated Raman studies in the supplementary information (Fig. 8) are referred to in the text, but no mention is made in the methods of the way in which the precursors are switched, growth times etc.

T. Booth

Version 1:

Reviewer comments:

Reviewer #1

(Remarks to the Author)

The authors reconstructed the manuscript with better logic, expanded the SAW-annealing discussion, enhanced the quality evaluation of 2D materials, and streamlined the oxidation content. The revised manuscript focuses more on the SAW-annealing technique and its corresponding application on 2D-material growth.

However, I still have some concerns:

1. In the revised MD simulations, the authors tracked the energy states and movement of Cu atoms under the traditional and SAW-assisted annealing, there truly exists a difference in the diffusion behavior of atoms. However, the time-scale difference is neglected: MD simulation is on the time scale of ns (roughness can reach 0.3~0.4 nm), but the experiments are with several to tens of minutes (roughness is still at the level of 1~2 nm). The data are not consistent. The authors should clarify them or provide more explanation. I noticed that the traditional annealing process also drives the Cu surface flatter. We are curious about whether the difference between "traditional treatment" and "SAW" can be narrowed over a longer period. Does the SAW-annealing solve the Cu foil flatten issue thermodynamically or kinetically?

2. As far as we know, the process of Cu (111) films on sapphire wafers has been well developed with high quality. The authors should highlight the advantages of SAW-annealing over Cu(111) wafer, such as the demonstration of flatness, the quality of resulting 2D materials, or the sample area (generally speaking, Cu foil has an advantage on large-scale preparation over wafer).

3. Fig. R10 should be included in supplementary information. Why is the E2g peak of hBN (at around 1370 cm⁻¹) absent in the last figure of Fig.R10 (g)?

4. Some minor suggestions:

---- In Fig. 1h, logarithmic coordinates are used for the horizontal axis (time axis), which should be marked in the figure legend.

----The optical images in Fig. 1e and Fig. S7d look out of focus. A small mark for focusing reference would make the data more convincing.

Reviewer #2

(Remarks to the Author)

The authors provided plenty of additional data and discussion, which greatly improved the quality and completeness of this work. The revisions to the article highlighted the innovations and most of the questions I raised before have been answered. However, there are still a few places that I hope the authors can answer and improve:

1. In Fig. 1f and Fig. 1h, the authors provide the time evolution of roughness in the experiment and in the MD simulation, but the time scales of the two are quite different, reaching 11 orders of magnitude, which casts doubt on the accuracy of this data. Can the authors provide explanation for this huge discrepancy?

2. In Fig. S1d, the surface roughness at 1070 °C has an increase that goes against common sense, as generally a higher temperature means a more intense thermal motion of atoms, making them easier to migrate to lower energy states. How does the author explain this phenomenon?

3. In Fig. 3c, the I_{2D}/I_G of graphene has a wide distribution from 1 to 5, which is not inconsistent with the authors' claims about the quality and uniformity of the graphene. The authors should provide further evidence, such as a large-area SEM photograph of this wafer and the I_D/I_G distribution diagram obtained from Fig. 3b.

4. Some minor issues: the illustration in Fig. 2f lacks height coordinates; the blue, red and white colors in Fig. 2b are not clearly marked and are confusing; and there is a ruler but no scale in Fig. 4b.

Version 2:

Reviewer comments:

Reviewer #1

(Remarks to the Author)

The authors have addressed all my concerns. The manuscript is ready for publication.

Reviewer #2

(Remarks to the Author)

The data and manuscript of the current version have been significantly improved, and I think it is now suitable for publication in Nature Communications.

Point-by-Point Responses to Reviewers' Comments

[Text in boldface = reviewers' comments; text in plain = authors' responses]

Reviewer #1:

This manuscript reports a surface acoustic wave (SAW) assisted annealing strategy to flatten Cu foil surfaces without polluting. Relying on the SAW-annealing technique, smooth and continuous graphene and hBN films are obtained.

The flatness of the substrate plays crucial roles in not only high-temperature growth but also post-growth peeling-transferring of 2D materials. Although ultra-flat Cu(111) and CuNi(111) wafers have been proposed via sputtering on sapphire (Nano Lett. 2020, 20 (9), 6798-6806) or CMP process (Adv. Mater. 2015, 27 (8), 1376-1382), mirror-like metal foils have not been reported so far. The pollution-free treatment of transforming commercial rough metal foils into ultra-flat substrates is an important issue. The SAW-annealing technique seems to pave the way for CVD-compatible ultra-flat growth of 2D materials.

Response:

We sincerely appreciate the reviewer's time and recognition of the novelty of our work. We are also grateful for the insightful suggestion regarding the significance of our SAW-annealing technique compared to sputtering methods for Cu films. In response, we have updated the introduction to discuss relevant research on sputtered metal films while emphasizing the unique contributions of our study. We believe these revisions better position our findings within the existing literature and underscore the crucial role of the SAW-annealing technique. The revised text now reads as follows:

“... **The flatness of the substrate plays a vital role not only in high-temperature growth processes but also in the subsequent transfer of 2D materials. ... While ultra-flat Cu(111) and CuNi(111) film wafers have been successfully fabricated through sputtering on sapphire substrates or via chemical mechanical polishing processes^{19,20}, the development of flat metal foils has yet to be reported. ...**”

*** Revised Manuscript P3 lines 12–18.**

“... ”

19 Deng, B. et al. Growth of ultraflat graphene with greatly enhanced mechanical properties. Nano Lett. 20, 6798-6806 (2020).

20 Nguyen, V. L. et al. Seamless stitching of graphene domains on polished copper (111) foil. Adv. Mater. 27, 1376-1382 (2015).

“... ”

*** Revised Manuscript P17 line 40–43.**

However, the authors paid much attention to the oxidation behaviour of graphene and hBN-coated Cu foils, including natural oxidation (Fig. 2), water-assisted oxidation (Fig. 3), and interfacial oxidation (Fig. 4). The DFT calculation and corresponding theoretical analysis are also about oxidation mechanism (Fig. 5). These extensive characterizations and analyses on Cu oxidation have crowded out the descriptions of SAW-annealing technique and the ultra-flat growth of high-quality graphene and hBN. Experimental details and mechanisms of SAW-annealing are unclear, and the quality evaluations on the resultant graphene and hBN are very limited. The title is no longer applicable to the content of the manuscript. I strongly recommend the authors reorganize the manuscript and put more space for SAW-annealing and ultra-flat growth of 2D materials.

Response:

We appreciate the reviewer for the valuable suggestions. We agree with the reviewer that the previous manuscript placed too much emphasis on the oxidation behavior of graphene and hBN-coated Cu foils, leading to insufficient focus on the SAW-annealing technique and the growth of high-quality graphene and hBN. To address these concerns, we have made the following revisions:

(1) Manuscript Restructuring: Under the reviewer's suggestions, we have reorganized the manuscript to better align with the title and the intended focus on SAW-annealing and ultra-flat growth of 2D materials. The revised structure now includes the following main sections: (1) SAW flattening strategy; (2) Flat Cu(111) foil via SAW annealing; (3) 2D material growth on flat Cu(111); and (4) Anticorrosion properties of hBN-coated Cu(111).

(2) Expanded SAW-Annealing Discussion: We have expanded the sections related to SAW-annealing, providing a detailed explanation of the experimental setup, parameters, and explanation of the mechanisms. This included a more in-depth exploration of the effects of SAW on atomic-level surface modification and the resulting improvements in material quality.

(3) Enhanced Quality Evaluation of 2D Materials: We included more additional experimental results and analyses that evaluate the quality of the as-grown graphene and hBN produced on the SAW-annealing treated Cu foils. This involves additional characterizations such as Raman spectroscopy, AFM, TEM, and others to demonstrate the superiority of the materials grown with SAW treatment.

(4) Streamlined Oxidation Content: In line with the reviewer's suggestion, we have streamlined the oxidation-related sections to ensure they support, rather than overshadow, the main narrative. The discussion on oxidation has been condensed and relocated to the latter part of the manuscript, emphasizing the protective capabilities of the SAW-treated surfaces and 2D materials as a complementary study.

These revisions ensure that the manuscript aligns more closely with its intended focus on the SAW-annealing technique and the high-quality growth of 2D materials. We believe these changes address the reviewer's concerns and improve the overall clarity and coherence of the manuscript. We thank the reviewer again for these insightful comments, which have greatly contributed to improving our work.

A list of the urgent issues follows below:

> **Comment 1.**

The sentence "External SAW facilitates the release of Cu atoms in higher energy states, enabling them to relocalize to more energetically favorable states" seems to explain the mechanism of the SAW strategy. However, no actual evidence has been presented, and the MD simulations in Supplementary Fig.2 only exhibited the result of SAW treatment. The process of the SAW in Fig,1a is not scientific enough. More details about SAW strategy should be proposed (e.g. energy states of Cu atoms?).

Response:

We appreciate the reviewer's insightful comment. We acknowledge that our initial explanation of the SAW mechanism was too simplistic and lacked sufficient technical detail. In response to the reviewer's suggestion, we have reconducted the molecular dynamics (MD) simulations to calculate the energy states of Cu atoms at various stages of the SAW process (Figure R1). We compared these states with those observed during high-temperature annealing without SAW, across different treatment durations. These simulations provide a more detailed understanding of how SAWs influence atomic movements and facilitate surface flattening. Specifically, we observed that external SAWs induce energy fluctuations among surface atoms, enabling them to relocate from high-energy, unstable sites to more stable, low-energy configurations, resulting in a smoother surface.

Figure R1. MD simulation of the Cu energy during the SAW process. a-b, Snapshots from MD simulations at different time points for traditional annealing at 1323 K (a) and SAW-assisted annealing at 1323 K (b). The average potential energy of Cu atoms was calculated and color-coded, ranging from -3.5 eV to -3.0 eV. Detailed atom energy distributions are highlighted for 1.5, 2.5, and 3.5 ns to allow for clearer observations.

We have revised the manuscript to include a more detailed explanation of the SAW mechanism, particularly focusing on the dynamic effects of SAWs on Cu atomic energy states, as observed in our MD simulations. We believe these revisions address your concerns and significantly enhance the manuscript's technical depth and clarity.

All these modifications have been included in the revised manuscript and supplementary information, and are shown here for your convenience:

“... ”

To further understand the underlying mechanism by which the SAWs reduce surface roughness during annealing, we performed MD simulations that tracked the energy states and movement of Cu atoms under the traditional and SAW-assisted annealing (Fig. 1g and Supplementary Fig. 2). The energy distribution revealed that most Cu atoms, except those on the surface, exhibited low potential energy, while surface atoms had higher potential energy due to reduced coordination and increased instability. As the system was heated, the crystal lattice began to degrade, leading to an increase in the potential energy of more atoms, as evidenced by the emergence of atoms with intermediate energy levels. The introduction of external vibrations through SAW treatment further enhanced this reconfiguration process. These vibrations induced additional energy fluctuations, particularly among surface atoms, facilitating their movement from high-energy, unstable sites to more stable, low-energy configurations. This increased atomic mobility led to a more significant reduction in surface roughness, producing a smoother and lower-energy surface compared to traditional annealing alone (Fig. 1h). These findings indicate that SAWs are an effective tool for manipulating atomic energy states and surface morphology, providing a more controlled and efficient method for achieving smooth Cu foils.

...”

* Revised Manuscript P5 line 19 – P6 line 10.

Fig. 1 | SAW-annealing treatments. ... g, MD simulations of the Cu surface during SAW annealing at 1323 K. The average potential energy of Cu atoms was calculated and color-coded. h, Comparison of surface roughness during traditional annealing (blue) and SAW annealing (red). ...”

*** Revised Manuscript P13 Figure 1.**

Fig. 2 | MD simulations of a Cu surface. a-b, Snapshots from MD simulations at different time points for traditional annealing at 1323 K (a) and SAW-assisted annealing at 1323 K (b). The average potential energy of Cu atoms was calculated and color-coded, ranging from -3.5 eV to -3.0 eV. Detailed atom energy distributions are highlighted for 1.5, 2.5, and 3.5 ns to allow for clearer observations. ...”

*** Revised Supplementary Information P3.**

> Comment 2.

It is proverbial that normal annealing and single-crystallization processes can also improve the flatness of the substrate (ACS Nano 2022, 16 (1), 285-294.), the authors should give more tangible evidence about the superiority of SAW compared with traditional annealing treatment and detailed process so that peers can evaluate the reproducibility of the experiment.

Response:

Thank you for your valuable suggestion. We agree with the reviewer that normal annealing and single-crystallization processes can indeed improve substrate flatness, as reported in *ACS Nano* 2022, 16(1), 285-294. Following the reviewer’s suggestion, we carefully compared traditional annealing with SAW-assisted annealing using SEM, optical, and AFM measurements. These comparisons are shown in the figures below (Figure R2, copied from Supplementary Fig. 5, and Figure R3, newly added as Supplementary Fig. 7 in this revision). The results reveal that while conventional annealing can reduce the surface roughness of commercial Cu foils from ~30 nm to ~4 nm (approximately a 6-fold reduction), the introduction of SAW further reduces

the roughness to ~ 1 nm (an additional 4-fold reduction). This enhanced flatness provides an ideal substrate for the growth of 2D materials and helps mitigate issues such as damage and large wrinkles caused by rough substrates.

Figure R2. AFM images of as-received Cu foils. a-c, AFM topography (a), amplitude error (b), and 3D map (c) images of non-treated Cu foil in the measurement area of $10 \times 10 \mu\text{m}^2$. The surface roughness was calculated as 30.8 nm. d, Line profiles for non-treated Cu surface extracted from AFM maps in (a).

Figure R3. Comparison of traditional and SAW-assisted annealing of Cu foils. a-d, Large-area SEM and optical images of Cu foils annealed using the traditional method (a, b) and the SAW-assisted annealing method (c, d). e-h, AFM images of Cu foils subjected to traditional annealing (e, f) and SAW-assisted annealing (g, h). i-j, AFM 3D map images of Cu foils after traditional annealing (i) and SAW-assisted annealing (j), with surface roughness measured at 3.66 nm and 0.27 nm, respectively, over a $1.0 \times 1.0 \mu\text{m}^2$ area. k-l, Line profiles of Cu surfaces from traditional annealing and SAW-assisted annealing, extracted from the AFM maps in (f) and (h).

In this revision, we have cited the mentioned work and included this comparison as a new section in the revised supplementary information. The corresponding description has also been added to the revised manuscript as follows:

“... It has been reported that normal annealing and single-crystallization processes can also improve substrate flatness³⁵. Therefore, we compared the surface morphology of SAW-annealed Cu(111) foil with that of traditionally annealed foil and observed a significant reduction in surface roughness in the SAW-treated sample (Supplementary Fig. 7). ...”

* Revised Manuscript P7 lines 2–11.

“... ”

35 Sun, L. *et al.* Toward epitaxial growth of misorientation-free graphene on Cu (111) foils. *ACS Nano* **16**, 285-294 (2021).

“... ”

* Revised Manuscript P18 lines 27–28.

Additionally, as suggested by the reviewer, we have included a more detailed description of the SAW annealing procedure in the Methods section, covering the equipment setup, technical details, and parameters used. These additions are intended to enhance the reproducibility of the experiment. We appreciate your important suggestion and believe these revisions effectively address your concerns.

“... ”

SAW treatment of Cu foil

The SAW treatment process was conducted using a custom-designed SAW-CVD system that integrates a SAW generator with an automated CVD growth system. This system produces acoustic waves at a fixed frequency of 20 kHz, with adjustable power levels up to 700 W. The SAW generator is connected to a quartz boat or shelf via a quartz rod, with all components pre-fused in the glass workshop to ensure a stable and secure connection during high-temperature operations. The prepared Cu(111) foil was placed in the quartz boat or shelf, depending on the number of samples being treated. This assembly was then connected to the SAW generator as described. The sample was slid into the CVD system and heated to an annealing temperature ranging from 1000 °C to 1070 °C. This temperature range was carefully selected to induce a quasi-liquid phase on the Cu foil surface, promoting surface smoothing through SAW treatment. Once the target temperature was reached, the sample was maintained at this temperature for 30 minutes to achieve thermal equilibrium. Following this stabilization period, the SAW generator was activated to apply acoustic waves to the Cu foil while maintaining the high annealing temperature. The duration of the SAW treatment varied from 10 seconds to 60 minutes, depending on the specific experimental conditions. The frequency of the SAWs was kept constant at 20 kHz, with power adjustable from 50 to 700 W. These parameters were optimized through controlled experiments to balance energy input with the desired surface modification effects. After the SAW treatment, the generator was shut down, and the sample was rapidly removed from the high-temperature zone, allowing it to cool naturally to room temperature.

“... ”

* Revised Manuscript P20 lines 10–31.

> **Comment 3.**

Terms such as “high-quality 2D materials”, “large-area”, “high crystal quality”, “absence of folds”, and “adlayer-free” are present several times in this manuscript, but the available evidence is insufficient to prove it. The present characterization area is too small to show its advanced nature (in Fig.2b,c, Fig.2e,f). The authors should provide characterization results on a larger scale and compare with the result on single-crystal Cu(111) foils (Adv. Mater. 2019, e1903615), CuNi(111) foils (Nature 2021, 596 (7873), 519-524), Cu(111) wafers (Nano Lett. 2020, 20 (9), 6798-6806), and CuNi(111) wafers (Nature 2020, 577 (7789), 204-208).

Response:

Thank you for your insightful suggestion. We have to admit that the previous version of our manuscript allocated too much space to the discussion of interfacial oxidation, leaving limited room for the large-scale characterization of 2D materials. In response to the reviewer’s suggestion, we have restructured the manuscript to include a dedicated section that thoroughly examines the quality of the grown 2D materials.

In this revision, we have provided more detailed characterization results that demonstrate the large-area, high-quality growth of 2D materials on SAW-treated Cu foil. This includes large-area characterizations focusing on material quality, fold, and adlayer examination across wafer-scale samples. These results have been compared with those reported in the references mentioned by the reviewer to provide a comprehensive evaluation. Additionally, we have incorporated these important references into our discussion to better contextualize our findings within the broader field of 2D material growth. We hope these revisions address the reviewer’s concerns and adequately demonstrate the advanced nature of our work. All the modifications in this revision are as shown below:

“... ”

2D material growth on flat Cu(111)

The smooth surface of the Cu foil is essential for the chemical vapor deposition (CVD) growth of large-area high-quality 2D materials. We synthesized wafer-scale monolayer graphene film on SAW-produced flat Cu(111) surfaces (Fig. 3a). After transferring on commonly used 4-inch 300-nm-thick SiO₂/Si substrates, Raman spectra were collected from 400 predefined positions arranged in the 20 by 20 point array on the 5.5 cm × 5.5 cm graphene film (Fig. 3b). These spectra exhibit the characteristic G and 2D bands of graphene, with the absence of the evident D band, indicating the high quality of as-grown monolayer graphene film. Statistical analysis of the 2D FWHM and I_{2D}/I_G ratios confirms the adlayer-free nature and exceptional quality of the monolayer graphene across the entire film area (Fig. 3c). Raman maps of the I_D/I_G ratio and 2D FWHM, acquired from nine randomly selected locations

across the graphene film, demonstrate a uniform monolayer of high quality, free from noticeable folds, adlayers, and defects (Supplementary Fig. 10). These characterizations are consistent with previous observations of CVD-grown, high-quality, adlayer-free graphene films^{2,10,38,39}. ...”

* Revised Manuscript P8 line 17 – P9 line 7.

Fig. 3 | 2D-material growth on SAW-treated flat Cu foils. a, Photograph of graphene film grown on SAW-treated flat Cu(111) foil, subsequently transferred to a 4-inch SiO₂/Si wafer. **b,** Raman spectra collected from 400 positions (20 × 20 array) on the graphene film shown in (a). **c,** Distribution of 2D FWHM and I_{2D}/I_G values from the monolayer graphene film, collected at 400 positions across the entire sample. ...”

* Revised Manuscript P15 Figure 3.

Fig. 10 | Raman maps of graphene grown on flat Cu(111) foil and transferred onto a SiO₂/Si substrate. a-i, I_{2D}/I_G ratio and 2D FWHM Raman maps collected from nine randomly selected locations across the entire graphene film on the SiO₂/Si wafer. The statistical FWHM values extracted from each map are presented in the lower panel. ...”

* Revised Supplementary Information P11.

> **Comment 4.**

More characterizations should be provided to prove the surface uniformity of the whole foils. At least several specified regions should be measured by AFM to give statistical results and to avoid accidental errors.

Response:

Thanks for the reviewer's comments and suggestions. We agree with the reviewer that characterizing individual regions alone does not fully represent the uniformity of the entire foil. In response to the reviewer's comments, we have carefully examined the SAW-treated Cu foil surface and conducted additional AFM measurements across several randomly selected regions to provide statistical results and minimize the possibility of accidental errors. These results have been added as a new section, as Figure 6 in the revised supplementary information.

Figure R4. Surface roughness measurements of SAW-treated Cu(111) foils. a-f, AFM topography (left) and phase (right) maps of SAW-annealed flat Cu foils from six randomly selected locations, with each measurement area covering $10 \times 10 \mu\text{m}^2$. The average surface roughness was calculated to be 0.99 nm. Line profiles extracted from each AFM topography map are shown below.

As shown in the above figure (Figure R4), the AFM maps collected from different locations on the foils demonstrate the uniform and flat surface of the prepared Cu foils via the SWA-assisted annealing method, validating the overall metal surface uniformity. We appreciate the reviewer's important suggestion and hope these additional results can well address your concerns.

> Comment 5.

In the DFT calculations, the authors froze all of the Cu atoms. To give expression to possible surface reconstruction and surface tension, Cu atoms on the surface should be relaxed sufficiently. In addition, the cutoff energy of 400 eV needs to be tested to obtain reliable results for this system with atoms such as N, O, and so on.

Response:

Thank you for your valuable suggestion.

Following the reviewer's recommendation, we have revised our DFT calculations with the following adjustments:

1. Relaxation of Surface Cu Atoms

We updated all calculations to allow for the relaxation of surface Cu atoms, which were previously kept frozen. While this adjustment led to a slight shift in the surface Cu layer and some reasonable changes in energy values, the overall trends remained consistent. This modification has enabled us to capture surface reconstructions and surface tension effects more accurately. The newly updated structures and calculated energies are included in the revised manuscript as follows:

Fig. 4 | Anticorrosion properties of monolayer hBN on flat Cu(111) foil. i, Top and side views of the atomic structures used for studying H₂O adsorption on bare flat (top) and hBN-covered flat (bottom) Cu surfaces. The Cu, B, N, O, and H atoms are shown in cyan, blue, magenta, red, and green, respectively. j, Adsorption energies of H₂O on the different Cu surfaces. ...”

* Revised Manuscript P16 Figure 4.

“... The Cu atoms are positioned in a face-centered cubic bulk structure, with all surface Cu atoms relaxed sufficiently. ...”

* Revised Manuscript P22 lines 40–42.

2. Cutoff Energy Testing

We conducted a convergence test of the total energy over an ENCUT range of 300 to 600 eV with increments of 50 eV. Our results indicated that the total energy difference reduced significantly to 0.26 meV/atom between 450 and 500 eV. Based on this test, we have revised all calculations using a 500 eV cutoff to ensure both accuracy and computational efficiency in this revision. These adjustments have been made to enhance the reliability and accuracy of our results. We appreciate your important suggestion and believe these changes could address your concerns. The modifications in the revised manuscript are shown below:

“... An energy cutoff of 500 eV was used for the plane wave expansion. ...”

* Revised Manuscript P22 lines 36–37.

> Comment 6.

The parameters of theoretical calculation should be clarified. For example, the SAW information of MD calculation needs to be described. Moreover, the supercell size and k-point mesh of DFT calculation are necessary to be mentioned.

Response:

We appreciate this valuable comment. Following the reviewer’s suggestion, we have added a more detailed description of the parameters used in our theoretical calculations to ensure clarity and reproducibility. Specifically as follows:

1. SAW parameters in MD calculations: We have provided a detailed description of the SAW parameters used in our MD simulations. This includes the frequency, amplitude, and duration of the SAW, as well as the specific interactions between the SAW and the Cu surface during the annealing process. All the modifications have been incorporated into the method section of the revised manuscript, as shown below:

“... Timesteps for the MD simulations were set to 0.1 femtoseconds. Initially, the system was equilibrated at 298 K for 1 ns. Following this, the temperature was ramped from 298 K to 1323 K linearly over 0.5 ns using a Nosé-Hoover thermostat. Upon reaching the peak temperature, the system was equilibrated for an additional 2 ns, both with and without external SAW. For the SAW simulation, the frequency and amplitude were set to 0.2 THz and 0.1 Å, respectively. Although in experiments, the external vibration has a much lower frequency (~20 kHz) and a larger amplitude (~100 μm), these parameters were adjusted to account for the limitations in length and time scales inherent to MD simulations. The system was then cooled down to 298 K over 0.5 ns and allowed to relax further at this temperature for 0.5 ns to achieve final surfaces. Throughout the simulation, the potential energy of each atom was averaged every 1000 steps, and these values were used to generate a color-coded atomic representation. The roughness was monitored throughout the simulation by calculating the average surface depth. ...”

* Revised Manuscript P22 lines 22–33.

2. Additional Details on DFT Calculations: Following the reviewer’s suggestion, we have provided a detailed description of the supercell dimensions and k-point mesh used in our DFT calculations. The selection of these parameters is critical for accurately capturing the electronic structure and surface phenomena. The modifications are listed as follows:

“... A supercell with lattice parameters $62.27 \times 2.57 \text{ \AA}$ was used for the flat Cu foil. A k-mesh of $1 \times 13 \times 1$ was employed for all these calculations. ...”

* **Revised Manuscript P22 lines 38–39.**

All these additional details have been incorporated into the revised manuscript to provide a clearer understanding of the computational setup and to effectively address the reviewer’s concerns.

We appreciate the time and valuable comments you have provided, which have significantly contributed to improving our manuscript. We hope that our revisions have effectively addressed your concerns and that our manuscript now meets the standards for publication.

Reviewer #2:

The author innovatively applied the SAW method to promote the movement of Cu atoms at high temperatures to achieve the flattening of Cu foils. As a demonstration of the application, the authors grew hBN on the ultraflat Cu foils and found that it had good anticorrosion properties. The author solves the long-standing problem of uneven substrates in large-scale 2D material production and avoids the introduction of additional impurities, which is of high significance.

Response:

We are very grateful to the reviewer for taking the time to review our manuscript and for recognizing the significance of our work. In this revision, we have carefully addressed all the comments and concerns raised, as outlined below.

I have the following comments, hoping to make this article better comply with the publishing standards of Nature Communications:

> Comment 1.

The most important thing is that the innovation of this article is that it proposes a new SAW treating method, but the discussion of the SAW treating process itself is very short. How do the temperature during SAW treatment, the length of treating time, the frequency and power of SAW, etc. affect the treating effect? What are the optimized specific parameters? The authors should provide the corresponding experiments in detail, and the method section should also be supplemented.

Response:

Thank you for your comments and suggestions. We fully agree with the reviewer that the previous version of our manuscript provided only a brief discussion on the SAW treatment, with more emphasis placed on interfacial oxidation. In response to the reviewer's suggestion, we have restructured the manuscript to include a more comprehensive discussion on the principles and detailed process of SAW treatment, supported by additional experimental results.

In this revision, we have expanded the discussion to address the effects of various SAW treatment parameters, including treatment duration (ranging from 0 to 60 minutes), SAW power (ranging from 50 W to 700 W), and annealing temperature (ranging from 1000 °C to 1070 °C).

The frequency of the SAW generator in our system is fixed at 20 kHz and cannot be adjusted. All this new information has been incorporated into Fig. 1 of the revised manuscript and Supplementary Fig.1 in the revised supplementary information, and for your convenience, it is also copied here as Figure R5. Through systematic optimization of these parameters, we identified the optimal conditions for producing a smooth Cu surface.

Figure R5. SAW treatment experiments. a-c, Surface roughness measurements of the treated Cu foil under varying SAW treatment time (10 s, 2 min, 10 min, 30 min, and 60 min), power levels (50 W, 100 W, 300 W, 500 W, and 700 W) and different annealing temperatures (1000 $^{\circ}\text{C}$, 1030 $^{\circ}\text{C}$, 1050 $^{\circ}\text{C}$, and 1070 $^{\circ}\text{C}$).

The detailed revision in the main manuscript is listed as follows:

“... The Cu foils were suspended on a custom-designed quartz shelf connected to a SAW generator, which was installed inside the quartz tube of the CVD system (Fig. 1b). The foils were then loaded into the high-temperature zone and heated to the desired annealing temperature. The SAW generator, operating at 20 kHz, was activated, transmitting acoustic waves to the samples through a quartz rod under high-temperature conditions. After SAW-annealing treatment, the surface of the Cu foils became significantly smoother compared to both the as-received and traditionally annealed Cu foils (Fig. 1c-e). Optimizing the SAW treatment parameters was crucial for achieving a smooth Cu surface. To determine the optimal conditions, a series of controlled experiments were conducted, focusing on treatment duration, SAW power, and annealing temperature (Fig. 1f and Supplementary Fig. 1). In our study, we employed specific conditions with an annealing temperature of 1323 K, a treatment duration of 30 minutes, and a SAW power of 500 W. These parameters resulted in optimal surface smoothness of the Cu foil while minimizing excessive surface evaporation. ...”

* Revised Manuscript P4 line 22 – P5 line 12.

Additionally, under the reviewer’s suggestion, we have also included a detailed discussion of the parameter optimization experiments, along with comprehensive technical information about the SAW system and instrumentation in the Methods section of the revised manuscript to enhance the reproducibility of our experiments. We appreciate the reviewer’s insightful comments and hope these additions provide a clearer understanding of the SAW treatment process and its effects on Cu foil.

The modifications are listed below:

“... **SAW treatment of Cu foil**

The SAW treatment process was conducted using a custom-designed SAW-CVD system that integrates a SAW generator with an automated CVD growth system. This system produces acoustic waves at a fixed frequency of 20 kHz, with adjustable power levels up to 700 W. The SAW generator is connected

to a quartz boat or shelf via a quartz rod, with all components pre-fused in the glass workshop to ensure a stable and secure connection during high-temperature operations. The prepared Cu(111) foil was placed in the quartz boat or shelf, depending on the number of samples being treated. This assembly was then connected to the SAW generator as described. The sample was slid into the CVD system and heated to an annealing temperature ranging from 1000 °C to 1070 °C. This temperature range was carefully selected to induce a quasi-liquid phase on the Cu foil surface, promoting surface smoothing through SAW treatment. Once the target temperature was reached, the sample was maintained at this temperature for 30 minutes to achieve thermal equilibrium. Following this stabilization period, the SAW generator was activated to apply acoustic waves to the Cu foil while maintaining the high annealing temperature. The duration of the SAW treatment varied from 10 seconds to 60 minutes, depending on the specific experimental conditions. The frequency of the SAWs was kept constant at 20 kHz, with power adjustable from 50 to 700 W. These parameters were optimized through controlled experiments to balance energy input with the desired surface modification effects. After the SAW treatment, the generator was shut down, and the sample was rapidly removed from the high-temperature zone, allowing it to cool naturally to room temperature. ...”

* Revised Manuscript P20 lines 10–31.

> **Comment 2.**

The author compares the impurity content of Cu foil after ECP treatment with that of Cu foil after high-temperature annealing, which is unfair. To grow 2D materials, the treated Cu foil will be subjected to high temperature for CVD growth, and many impurities may volatilize or react during the high-temperature process. Will the Cu foil that is ECP treated first and then high-temperature annealed have a flatter surface and a lower or similar impurity content?

Response:

We appreciate the reviewer’s insightful comment, which highlights an important consideration when comparing Cu foils treated by ECP with those treated by SAW, particularly regarding the effects of subsequent high-temperature processes. We agree that directly comparing Cu foils treated solely by ECP with those treated by SAW may not be entirely fair if the impact of high-temperature annealing is not taken into account.

In fact, we were aware of literature reporting that annealing after ECP treatment can effectively reduce surface contamination (please kindly refer to Advanced Materials 34.8 (2022): 2108608). Therefore, in our experiments, we included a post-annealing step to ensure a fair comparison. Specifically, after ECP treatment, the Cu foils were annealed under H₂ (50 sccm) and Ar (50 sccm) for 1 hour in our CVD system. We then compared the impurity content of these annealed samples with those treated by SAW using XPS and D-SIMS analysis.

Additionally, from our XPS and D-SIMS results, we observed a noticeable increase in phosphorus (P) levels in the ECP-treated Cu after annealing (Fig. 2i-j in the manuscript and reproduced here as Figure R6). However, there was no significant rise in other contaminants such as sulfur (S) or nitrogen (N) in the ECP-treated Cu foils (Supplementary Fig. 8 and 9 in the supplementary information and reproduced here as Figures R7 and R8). This finding aligns well with the report in *Advanced Materials* 34.8 (2022): 2108608, where post-annealing was shown to remove most contaminants, with P persisting inside the Cu foils, as described:

“... Moreover, the impurity concentrations of S, Cl, and Si elements show a clear decreasing trend with the cycle number, strongly indicating that the impurities are removed by the treatment (Figure S9, Supporting Information). Note that the impurity of P element essentially remains the same with a small fluctuation, possibly due to the fact that the electrochemical polishing solution contains phosphoric acid (Figure S8, Supporting Information). ...”
 – Ref: *Advanced Materials* 34.8 (2022): 2108608

Figure R6. Phosphorus contaminants measurements. i, XPS spectra of P 2p core level collected from the non-treated (pink), SAW-treated (blue), and ECP-treated (green) Cu foils. j, D-SIMS spectra of the P contaminant intensity in three types of Cu foils.

Figure R7. XPS spectra of potential contaminants in Cu foils. a-c, XPS spectra of S 2p core level collected from the non-treated (a), ECP-treated (b), and SAW-treated (c) Cu foils. d-f, XPS spectra of N 1s core level collected from the non-treated (d), ECP-treated (e), and SAW-treated (f) Cu foils.

Figure R8. SIMS measurements of Cu foils. a-b, D-SIMS spectra of nitrogen contaminate intensity inside three types of Cu foils. c-d, D-SIMS spectra of sulfur contaminate intensity inside three types of Cu foils.

To further clarify this in our manuscript, we have revised the main text and Methods section to provide a clearer and more accurate comparison. We apologize for not making this explicit in the previous version and appreciate the reviewer’s question, which has helped improve the clarity of our work.

All modifications are listed below:

“... It has been reported that post-annealing can remove some chemical contamination from ECP-treated Cu foil³⁷. Therefore, to ensure a fair comparison in our study, the ECP-treated Cu foils were annealed prior to the XPS measurements (see Methods). ...”

* Revised Manuscript P7 lines 19–21.

“... For a fair comparison, we annealed the Cu foil in the CVD system under H₂ (50 sccm) and Ar (50 sccm) for 1 hour following the ECP treatment. ...”

* Revised Manuscript P20 lines 38–39.

“... ”

37 Yao, W. et al. Bottom-Up-Etching-Mediated Synthesis of Large-Scale Pure Monolayer Graphene on Cyclic-Polishing-Annealed Cu (111). *Adv. Mater.* 34, 2108608 (2022).

“... ”

* Revised Manuscript P18 lines 31–33.

> **Comment 3.**

In the analysis in Fig. 5 and the schematic diagram in Fig. S1, the authors implied that the SAW-treated Cu foil has only 1-atom steps. More experiments are needed to prove this, including atomically resolved cross-sectional STEM images.

Response:

We thank the reviewer for highlighting this important issue.

We acknowledge that our previous description was inaccurate, and we appreciate the reviewer pointing out this issue. In reality, our experimental observations show that the SAW-treated Cu foil does not consist of single-atom steps; instead, it exhibits several atomic-height steps.

To address this more accurately, as suggested by the reviewer, we prepared cross-sectional TEM samples from the SAW-treated flat Cu(111) foils using FIB and acquired atomically resolved cross-sectional HAADF-STEM images in different regions. As shown in Figure R9a, the Cu(111) surface is atomically flat in most areas. However, upon systematic inspection, we found that in some regions, the Cu surface still exhibits steps with several atomic heights, as revealed in the atomically resolved STEM image in Figure R9b. Furthermore, we conducted low-temperature STM measurements at the step regions, which confirmed step heights similar to those observed in the HAADF-STEM images.

Figure R9. Atomically resolved characterizations of the surface steps of the SAW-treated Cu foils. a, Cross-sectional HAADF-STEM image of the surface region of the produced flat Cu(111) foil acquired from the flat region. b, Cross-sectional HAADF-STEM image acquired from the step region. c, Low-temperature STM image of the Cu surface at the step edge region, with the line profile across the step shown below.

We must admit that although the surface roughness has been significantly reduced through SAW treatment compared to the as-received Cu foil, the surface is not perfectly atomically flat. Some surface steps with several atomic heights remain.

We appreciate the reviewer's question, as it is crucial to provide an accurate description. In the revised manuscript, we have removed the misleading schematic diagram, included the newly obtained STEM images, and clarified the description to explicitly state that the SAW-treated Cu foil surface exhibits few-atom steps rather than perfect single-atom steps. We hope these revisions and clarifications address your concerns and accurately reflect our experimental findings.

All modifications are listed below:

“... Moreover, we prepared cross-sectional lamellae using focused ion beam (FIB) to further examine the surface of the SAW-treated Cu(111) foils. High-angle annular dark-field scanning transmission electron microscopy (HAADF-STEM) images revealed a generally flat surface on the produced Cu(111) foil (Fig. 2g). However, we observed that the surface is not perfectly atomically flat everywhere; in certain regions, steps with heights equivalent to several atoms are still present, as depicted in Fig. 2h. ...”

* Revised Manuscript P7 lines 6–11.

> **Comment 4.**

According to the literature, flat Cu foil can prevent corrosion itself (ref. 8). Can the authors compare the anticorrosion effects of rough Cu, flat Cu, and flat Cu with hBN?

Response:

We appreciate this valuable suggestion from the reviewer. We also noted that the study in Ref. 8 suggests that ultraflat Cu(111) foils can inherently offer improved corrosion resistance itself compared to rough Cu surfaces. Following the reviewer's suggestion, we have conducted the compared experiments of three types of samples: rough Cu produced by traditional annealing, flat Cu produced by SAW treatment, and flat Cu covered with a monolayer of hBN.

The results are shown in the following figure (Figure R10). After the oxidation treatment, we observed that both bare rough and flat Cu foils oxidized, as evidenced by the detection of Cu_xO signals on their surfaces from the Raman measurements. In contrast, the hBN-covered flat Cu showed no signs of oxidation, demonstrating good anti-corrosion properties due to the protective nature of the hBN layer.

When comparing the rough and flat bare Cu foils, the rough foil appeared more heavily oxidized. However, our experimental results differ from those reported in the work of reference 8, where flat Cu foil was shown to effectively prevent oxidation. We analyzed these results and

suggested that this discrepancy may be due to the fact that our SAW-produced Cu foil is not perfectly flat; as mentioned in a previous response, there are still some surface steps with several atomic heights. This differs from the sputter-deposited Cu foils in the referenced work, which have remarkably atomically flat surfaces. These residual surface steps could be the reason why oxidation was still observed on our flat sample.

Figure R10. Oxidation experiments on rough and flat Cu foils. **a**, Photograph of rough Cu foil, SAW-treated flat Cu foil, and flat Cu foils covered with a monolayer of hBN after undergoing oxidation treatment. **b-d**, Optical images acquired from rough Cu foil, SAW-treated flat Cu foil, and flat Cu foils with monolayer hBN coverage. Typical Raman spectra for each of these samples are displayed below.

> **Comment 5.**

According to the literature, bilayer graphene can solve its electrochemical corrosion problem (Nature Communications, 2023, 14(1): 7447). As the authors claimed that the oxidation at the edges of graphene islands is induced by galvanic corrosion, would bilayer graphene have a better performance?

Response:

Thank you for this comment. We also noticed the findings of Prof. Liu et al. (*Nature Communications*, 2023, 14(1): 7447) regarding the superior anticorrosion performance of bilayer graphene. The improved corrosion resistance in bilayer or multilayer graphene is attributed to the ability of multiple layers to better shield the underlying graphene from oxygen, thereby mitigating electrochemical corrosion at the edges of graphene islands.

The additional graphene layers provide better protection by blocking oxygen from reaching the layers below, which reduces exposure and effectively prevents galvanic corrosion. Therefore, we agree with the reviewer that, as reported in the literature, bilayer graphene is expected to offer improved performance due to this interlayer protection effect. We have incorporated this discussion and cited the referenced work in the revised manuscript to provide a more comprehensive understanding of the anti-corrosion capabilities of graphene with different layer configurations. In future work, we also plan to investigate the influence of 2D-material layers on the anti-oxidation properties of metal foils.

All modifications are listed below:

“... It has been reported that bilayer graphene can solve the electrochemical corrosion problem and enhance anti-corrosion properties by blocking water and oxygen from reaching the Cu surface⁴³. ...”

* Revised Manuscript P10 line 22 – P11 line 1.

“... 43 Zhao, M. *et al.* Enhanced copper anticorrosion from Janus-doped bilayer graphene. *Nat. Commun.* 14, 7447 (2023). ...”

* Revised Manuscript P18 lines 44–45.

> **Comment 6.**

Some minor problems, the font size of Fig. 2d is too small; Fig. 3a lacks the corresponding labels and illustrations of each part.

Response:

We appreciate the reviewer for pointing out these issues. We have adjusted the font size in Fig. 2d (now Fig. 3g in the revised manuscript) to ensure it is legible. Additionally, we have added the necessary labels and illustrations to Fig. 3a (now Fig. 4a in the revised manuscript) and have carefully reviewed all figures to address any related concerns.

We greatly appreciate the reviewer’s time and constructive suggestions, which have been invaluable in improving our manuscript. We hope that we have fully addressed the reviewer’s concerns and that the manuscript is now suitable for publication.

Reviewer #3:

The authors present results on the smoothing of catalytic copper foils during 2D materials synthesis in furnaces using a surface acoustic wave technique.

The authors demonstrate that the SAW treatment provides an additional annealing effect on the smoothness of the catalytic substrates, with supporting evidence from AFM and cross-sectional TEM, as well as Raman spectroscopic investigation of isotopically differentiated growths and EBSD crystal orientation investigations.

The authors also demonstrate that while the SAW results are comparable to the more widely employed electrochemical polishing techniques, there is no unwanted contamination introduced thanks to the lack of an electrolyte. They demonstrate this with XPS and D-SIMS spectroscopies.

Finally, the authors use the 2D materials layers in anticorrosion studies to determine the rate of diffusion of oxygen under flakes, used as a proxy for the strength of the surface binding, and supporting the work with Raman mapping and HAADF-STEM, HRTEM and EELS.

The experimental work is supported by DFT simulations of water binding to the Cu(111) surface, coated with hBN, coated with graphene over steps and bare.

Overall, the work is detailed, interesting and the conclusions are very well supported by the comprehensive experimental and simulation data. The technique presented shows strong benefits for the production of 2D materials on metal catalyst layers by CVD.

Response:

We greatly appreciate the reviewer's time and detailed evaluation of our work.

I would be delighted to recommend publication after the following issues are addressed:

> Comment 1.

Please provide much more method detail on the SAW device, coupling to the system, commercial vs. homebrew, power, time, etc. Details are almost completely absent from the manuscript, and the study as it stands would be impossible to replicate or verify.

Response:

Thank you for your valuable comment.

In response to the reviewer's suggestion, we have significantly expanded the Method section in the revised manuscript to include detailed information about the SAW device, its integration into the system, and the specific parameters used in our study. This includes the power levels, treatment duration, and coupling methods, which were not fully described in the previous version of the manuscript. We believe these additional technical details will enhance the reproducibility of our study and facilitate broader use of the SAW technique for smoothing Cu foils.

All the modifications are listed below:

“ ...

SAW treatment of Cu foil

The SAW treatment process was conducted using a custom-designed SAW-CVD system that integrates a SAW generator with an automated CVD growth system. This system produces acoustic waves at a fixed frequency of 20 kHz, with adjustable power levels up to 700 W. The SAW generator is connected to a quartz boat or shelf via a quartz rod, with all components pre-fused in the glass workshop to ensure a stable and secure connection during high-temperature operations. The prepared Cu(111) foil was placed in the quartz boat or shelf, depending on the number of samples being treated. This assembly was then connected to the SAW generator as described. The sample was slid into the CVD system and heated to an annealing temperature ranging from 1000 °C to 1070 °C. This temperature range was carefully selected to induce a quasi-liquid phase on the Cu foil surface, promoting surface smoothing through SAW treatment. Once the target temperature was reached, the sample was maintained at this temperature for 30 minutes to achieve thermal equilibrium. Following this stabilization period, the SAW generator was activated to apply acoustic waves to the Cu foil while maintaining the high annealing temperature. The duration of the SAW treatment varied from 10 seconds to 60 minutes, depending on the specific experimental conditions. The frequency of the SAWs was kept constant at 20 kHz, with power adjustable from 50 to 700 W. These parameters were optimized through controlled experiments to balance energy input with the desired surface modification effects. After the SAW treatment, the generator was shut down, and the sample was rapidly removed from the high-temperature zone, allowing it to cool naturally to room temperature.

...”

* Revised Manuscript P20 lines 10–31.

> Comment 2.

The isotopically differentiated Raman studies in the supplementary information (Fig. 8) are referred to in the text, but no mention is made in the methods of the way in which the precursors are switched, growth times etc.

Response:

Thank you for your suggestion and for pointing out this issue.

In response to the reviewer's comment, we have included additional details in the Methods section of the revised manuscript and supplementary information. These details cover the specific growth parameters, such as gas flow ratios, switching durations of carbon sources, growth temperatures, and other relevant conditions.

“... For the isotopic labeling graphene growth experiments, commonly used ^{12}C methane and ^{13}C isotopic labeling methane gases were used as carbon sources in a CVD growth under controlled conditions. During the graphene synthesis, 5 sccm of methane with either ^{13}C or ^{12}C was alternately introduced into the growth chamber, with each isotopic growth period lasting 10 minutes to allow sufficient time for graphene formation from each carbon source. This alternation between ^{13}C and ^{12}C was repeated multiple times to produce distinct isotopically labeled graphene regions. Throughout the process, the furnace temperature was kept at $1030\text{ }^\circ\text{C}$, and hydrogen was supplied at a constant flow rate of 20 sccm. After the growth process, the sample was slid into the room temperature zone. ...”

* Revised Manuscript P21 lines 8–16.

Fig. 11 | Graphene islands grown on Cu foil after natural oxidation. a, Schematic diagram of graphene island growth on flat Cu foil in a CVD system, showing the growth parameters, including gas flow ratios of $^{13}\text{C}\text{-CH}_4$, $^{12}\text{C}\text{-CH}_4$, H_2 , and growth temperature during the CVD growth process. **b-c**, Raman maps of the $^{12}\text{C}\text{-G}$ (b) and $^{13}\text{C}\text{-G}$ (c) band intensities. **d-e**, Raman maps of the $^{12}\text{C}\text{-2D}$ (d) and $^{13}\text{C}\text{-2D}$ (e) band intensities. **f-g**, Raman maps of 2D FWHM of ^{12}C (f) and ^{13}C (g) graphene regions. **h**, Raman intensity line profiles of Cu_xO and graphene-2D bands along the dashed line in Fig. 3g. ...”

* Revised Supplementary Materials P12.

We appreciate the reviewer's valuable suggestions, which have significantly contributed to improving our manuscript.

Point-by-Point Responses to Reviewers' Comments

[Text in boldface = reviewers' comments; text in plain = authors' responses]

Reviewer #1:

The authors reconstructed the manuscript with better logic, expanded the SAW-annealing discussion, enhanced the quality evaluation of 2D materials, and streamlined the oxidation content. The revised manuscript focuses more on the SAW-annealing technique and its corresponding application on 2D-material growth.

Response: We sincerely appreciate the reviewer's recognition of the revisions made in the previous round. Additionally, we are very grateful for the valuable suggestions provided, which helped us restructure the manuscript in a much improved manner. These suggestions have played a significant role in enhancing the quality of our paper.

However, I still have some concerns:

1. In the revised MD simulations, the authors tracked the energy states and movement of Cu atoms under the traditional and SAW-assisted annealing, there truly exists a difference in the diffusion behavior of atoms. However, the time-scale difference is neglected: MD simulation is on the time scale of ns (roughness can reach 0.3~0.4 nm), but the experiments are with several to tens of minutes (roughness is still at the level of 1~2 nm). The data are not consistent. The authors should clarify them or provide more explanation. I noticed that the traditional annealing process also drives the Cu surface flatter. We are curious about whether the difference between "traditional treatment" and "SAW" can be narrowed over a longer period. Does the SAW-annealing solve the Cu foil flatten issue thermodynamically or kinetically?

Response:

We sincerely thank the reviewer for raising this important issue. We agree that there is a significant time-scale difference between the MD simulations and the actual experimental observations. In fact, we also noticed this discrepancy during discussions with our theoretical team. Please let us to explain this difference from the following perspectives:

Computational Method Limitation

The MD simulations are inherently limited by significant computational challenges, which restrict simulations to very short time scales, typically in the picosecond (ps) to nanosecond (ns) range. These limitations arise due to the significant computational demands required to model interactions between thousands to millions of atoms, the complexity of interatomic potentials, and the extremely small time steps (on the order of femtoseconds) necessary to accurately capture atomic motions. In contrast, many real-world physical processes occur on much longer time scales, ranging from minutes to hours. Achieving such extended time scales in MD simulations is virtually impossible due to the enormous computational resources needed. For instance, to simulate just milliseconds of real-time molecular dynamics with atomistic precision, one would need billions of time steps, which would take an immense amount of computational power. While specialized hardware has extended simulations into the millisecond range until today, processes occurring over seconds or even longer (such as minutes and hours) remain far out of reach. Similar challenges have been discussed extensively in *Nature Communications 11.1 (2020): 2918*. Consequently, researchers often rely on shorter time scale simulations to approximate and describe the underlying physical mechanisms.

Widely Used for Exploring Atomic Mechanisms

Despite the inherent time-scale discrepancies between MD simulations and real-world processes, MD simulations remain widely used by researchers to observe atomic-scale changes in lattice structures and atomic dynamics. These simulations provide valuable insights into the evolution of physical processes at the atomic level. Although the time scales in MD simulations are significantly shorter than those in real experiments, they are still an essential tool for supporting experimental findings.

For instance, in *Physical Review Letters 120.24 (2018): 246101*, MD simulations with a time scale of approximately 3 nanoseconds (ns) were employed to study the surface step bunching effect on Cu surfaces during graphene growth. The simulations revealed the atomic mechanisms driving the step bunching, such as the rapid diffusion of metal adatoms beneath graphene layers. These results were later confirmed through corresponding experimental observations, validating the use of MD simulations despite the shorter time frame. This combination of theoretical and experimental approaches demonstrates how MD simulations, even on limited time scales, can effectively replicate and predict real-world physical phenomena.

Supporting Experimental Observations

Although the time scales of MD simulations and real-world processes differ significantly, in our study, MD simulations play a critical role in exploring the underlying atomic-level mechanisms and energy transformations that drive surface smoothing under the influence of SAW. These simulations provide insights into atomic rearrangements and energy dissipation that are challenging to observe directly in experiments, such as how SAW impacts atomic motion on the Cu surface to facilitate its flattening. Despite their shorter time scales, MD simulations offer essential theoretical support that helps to explain and validate our experimental findings.

Surface Roughness Difference

Regarding the discrepancy in surface roughness—where MD simulations yield values of 0.3–0.44 nm, while experimental results range from 1 to 2 nm, we suggest that this can be attributed to differences in sample area sizes. In our study, the MD simulation covers an area of tens of nanometers in length, while experimental measurements, such as those using AFM, typically span much larger areas (e.g., $10\ \mu\text{m} \times 10\ \mu\text{m}$). Larger sampling areas generally result in higher surface roughness values. Similarly, if we see the STM measurement results in Fig. 4h, where the surface roughness is $\text{RMS} = 0.402\ \text{nm}$ in a $20\ \text{nm} \times 20\ \text{nm}$ region, while in a smaller $5\ \text{nm} \times 5\ \text{nm}$ area, it is $\text{RMS} = 0.043\ \text{nm}$ —closely matching the MD simulation results.

Thermodynamically or Kinetically

Regarding the thermodynamic and kinetic considerations of SAW annealing: in principle, we agree with the reviewer's perspective that longer timescales could reduce the differences between traditional annealing and SAW. We even thought that increasing the temperature in traditional annealing could also help narrow the gap. However, in our experiments, we observed that longer annealing times or higher temperatures in traditional annealing often led to excessive Cu evaporation, resulting in a rougher surface. In contrast, using SAW as an additional energy input at a relatively lower annealing temperature provides extra thermal energy (a thermodynamic process) or effectively extends the annealing time, without causing excessive evaporation that contributes to surface roughness increase. We attribute this to the fact that SAW not only enhances the thermodynamic aspects of annealing but also plays a role in the kinetic process of surface atom redistribution. Thus, we suggest that both thermodynamic and kinetic mechanisms are involved in the SAW treatment. We will continue to further explore the mechanism in our future studies.

We sincerely appreciate the reviewer's suggestions. Following the reviewer's guidance, we have further clarified and expanded on the time scale differences in the revised manuscript. We hope that these modifications enhance the clarity and overall quality of the paper.

All modifications are listed as follows:

“... Although MD simulations are performed on much shorter time scales than real-world processes due to computational limitations, they effectively reveal the atomic mechanisms through which SAW plays a crucial role in facilitating the flattening of the Cu surface. ...”

* Revised Manuscript P6 lines 8–11.

2. As far as we know, the process of Cu (111) films on sapphire wafers has been well developed with high quality. The authors should highlight the advantages of SAW-annealing over Cu(111) wafer, such as the demonstration of flatness, the quality of resulting 2D materials, or the sample area (generally speaking, Cu foil has an advantage on large-scale preparation over wafer).

Response:

We appreciate the reviewer's insightful comment. We fully agree that the preparation of Cu(111) films on sapphire wafers is a well-established process and can indeed result in ultra-smooth surfaces. In line with the reviewer's suggestion, we have revised the manuscript to underscore the advantages of SAW-annealing on Cu foil, such as the potential for large-scale production and lower manufacturing costs. We are grateful for this suggestion, as it has allowed us to better emphasize the relevance and impact of our research.

The revisions are as follows:

“... While high-quality, ultra-flat Cu(111) and CuNi(111) film wafers have been successfully developed through sputtering on sapphire substrates or chemical mechanical polishing processes^{19,20}, developing efficient flattening methods for metal foils remains crucial for achieving cost-effective production and scalable preparation of 2D materials. ...”

* Revised Manuscript P3 lines 16–19.

3. Fig. R10 should be included in supplementary information. Why is the E_{2g} peak of hBN (at around 1370 cm⁻¹) absent in the last figure of Fig.R10 (g)?

Response:

Thank you for your valuable suggestion. Based on the reviewer's recommendation, we have now included Fig. R10 in the revised Supplementary Information as new Fig. S13.

Now, let us clarify the absence of the E_{2g} peak of hBN on Cu foil. It is important to note that the Raman signal of the hBN E_{2g} band is significantly weaker compared to other 2D materials like graphene and transition metal dichalcogenides. This is primarily due to the non-resonant Raman scattering in hBN, as discussed in *2D Materials 4.3 (2017): 031003*. Consequently, under the strong Raman fluorescence background of Cu, it is extremely difficult to see the weak Raman E_{2g} signal of hBN. This differs from graphene, which has strong Raman signals, making it easier to observe on Cu surfaces. For this reason, hBN is typically transferred onto non-metal substrates to detect its Raman signal. Therefore, in the Raman spectrum of Fig. R10(g), only the Cu fluorescence background is visible, and the hBN Raman peak is not detected. We hope this explanation adequately addresses your concern regarding the absence of the hBN E_{2g} peak.

4. Some minor suggestions:

---- In Fig. 1h, logarithmic coordinates are used for the horizontal axis (time axis), which should be marked in the figure legend.

----The optical images in Fig. 1e and Fig. S7d look out of focus. A small mark for focusing reference would make the data more convincing.

Response:

We thank the reviewer for the helpful suggestions.

Based on the reviewer's advice, we have updated the figure legend of Fig. 1h to indicate the use of logarithmic coordinates on the horizontal (time) axis. Additionally, following the reviewer's recommendation, we have replaced the optical images in Fig. 1e and Fig. S7d by selecting slightly imperfect areas or debris as a reference point for improved focus. The images now appear clearer and more convincing. We greatly appreciate the reviewer's valuable input, which has helped enhance the clarity and professionalism of our paper.

All modifications are listed here for your convenience:

“... **Fig. 1 ... h**, Comparison of surface roughness during traditional annealing (blue) and SAW annealing (red). The horizontal axis (time) is displayed on a logarithmic scale. ...”

* Revised Manuscript P13 Figure 1.

“... ”

“... ”

* Revised Manuscript P13 Figure 1.

“... ”

“... ”

* Revised Supplementary Information P7 Figure S7.

Reviewer #2:

The authors provided plenty of additional data and discussion, which greatly improved the quality and completeness of this work. The revisions to the article highlighted the innovations and most of the questions I raised before have been answered.

Response:

We appreciate the reviewer's kind words and recognition of our work. In this round of revisions, we have carefully addressed and clarified all remaining concerns raised by the reviewer. We hope these modifications enhance the manuscript and make it suitable for publication.

However, there are still a few places that I hope the authors can answer and improve:

1. In Fig. 1f and Fig. 1h, the authors provide the time evolution of roughness in the experiment and in the MD simulation, but the time scales of the two are quite different, reaching 11 orders of magnitude, which casts doubt on the accuracy of this data. Can the authors provide explanation for this huge discrepancy?

Response:

We sincerely thank the reviewer for raising this important question, which we have also noted as a concern from Reviewer #1. Here, please let us provide a clear explanation of the time scale discrepancy to avoid confusion among readers in this revision

We took this issue seriously and had an in-depth discussion with our theoretical calculation team. We believe that the large time scale difference between MD simulations and experimental processes is due to the inherent limitations of MD simulation methods. Typically, MD simulations are conducted on the nanosecond (ns) or picosecond (ps) scale, as extending them to minutes or hours would require immense computational resources, making it impractical. In contrast, real-world experiments often span tens of minutes or even hours, making it impossible for MD simulations to reach comparable time scales given the current computational capabilities (Kindly refer to more discussions in *Nature Communications* 11.1 (2020): 2918).

Despite these time scale discrepancies, MD simulations are still widely used by researchers to observe atomic-scale changes and dynamic mechanism. They can provide valuable insights

and exploration on the atomic-level physical processes, which effectively support experimental observations, even on very different time scales (an example is demonstrated in *Physical Review Letters* 120.24 (2018): 246101). MD simulations are essential for exploring underlying physical mechanisms and validating the feasibility of processes. Therefore, while simulations cannot match the time scales of real-world processes, they still provide essential theoretical support by revealing differences in atomic motion and surface roughness changes under the influence of SAW treatments.

Based on the reviewer's suggestion, we have added more detailed explanations to clarify this issue for readers. We sincerely thank the reviewer for these valuable suggestions, which have greatly enhanced the clarity of our paper.

All modifications are listed as follows:

“... Although MD simulations are performed on much shorter time scales than real-world processes due to computational limitations, they effectively reveal the atomic mechanisms through which SAW plays a crucial role in facilitating the flattening of the Cu surface. ...”

* Revised Manuscript P6 lines 8–11.

2. In Fig. S1d, the surface roughness at 1070 °C has an increase that goes against common sense, as generally a higher temperature means a more intense thermal motion of atoms, making them easier to migrate to lower energy states. How does the author explain this phenomenon?

Response:

We sincerely thank the reviewer for raising this insightful question. We, too, observed this unexpected phenomenon during our experiments. Initially, we hypothesized that higher temperatures would result in a smoother Cu surface due to enhanced atomic mobility, as one would expect atoms to migrate to lower energy states more easily. However, in the experiment, we found that after annealing at 1070°C with the SAW, the surface roughness increased. We repeated the experiments several times to confirm this unusual phenomenon. Upon careful analysis, we attributed this to the excessive evaporation of Cu surface atoms. The SAW process provides additional energy input (ΔT), and at high temperatures like 1070°C, prolonged annealing can roughly equate (though not fully) to heat treatment at even higher temperatures

(though not identical) to heat treatment at even higher temperatures. This excessive energy likely leads to over-evaporation of surface atoms, resulting in increased surface roughness rather than smoothing it.

In response to the reviewer's suggestion, we have added further explanations of this anomalous behavior in the revised manuscript. All changes have been listed here for your convenience.

“... Notably, while higher temperatures typically reduce surface roughness, our experiments showed an increase in roughness at 1070°C. We attribute this to excessive evaporation of Cu surface atoms, caused by the high annealing temperature combined with the additional energy input from the SAW process. ...”

* Revised Supplementary Information P2 Figure 1.

3. In Fig. 3c, the I_{2D}/I_G of graphene has a wide distribution from 1 to 5, which is not inconsistent with the authors' claims about the quality and uniformity of the graphene. The authors should provide further evidence, such as a large-area SEM photograph of this wafer and the I_D/I_G distribution diagram obtained from Fig. 3b.

Response:

We thank the reviewer for highlighting this important point. During our study, we also noticed this relatively wide distribution of I_{2D}/I_G values, with several values falling within the range of 1 to 2. Therefore, to further clarify, we additionally extracted the FWHM distribution, where all FWHM values are between 20 and 35, confirming that the as-grown graphene is a monolayer.

We appreciate the reviewer's suggestion to include the I_D/I_G distribution to better demonstrate the quality of the graphene. In this revision, under the reviewer's suggestion, we have replaced the I_{2D}/I_G plot with the I_D/I_G distribution, derived from 400 Raman spectra (see Fig. 3d and Figure R1). The absence of significant defect signals (with most I_D/I_G values below 0.3) confirms the high quality of as-grown graphene. Additionally, we have included a large-area SEM image of the as-grown graphene, which further illustrates the uniformity of the film.

All these revisions have been included into the revised manuscript and Supplementary Information to better clarify and substantiate the quality and uniformity of the graphene grown in our study.

“... ”

Fig. 3 ... c, Distribution of 2D FWHM and I_D/I_G ratios from the monolayer graphene film, collected at 400 positions across the entire sample. ...”

*** Revised Manuscript P15 Figure 3.**

“... ”

Fig. 10 | Large-scale SEM image of as-grown monolayer graphene on Cu(111) foil.

...”

*** Revised Supplementary Information P11 Figure 10.**

4. Some minor issues: the illustration in Fig. 2f lacks height coordinates; the blue, red and white colors in Fig. 2b are not clearly marked and are confusing; and there is a ruler but no scale in Fig. 4b.

Response:

We appreciate the reviewer for pointing out these minor issues in our manuscript.

Following the reviewer’s suggestions, we have added the height coordinates to Fig. 2f. Regarding Fig. 3b, the blue, red, and white colors of the spectra were originally used to distinguish the stacked 400 spectra lines for clearer illustration, but they do not have any scientific significance. We apologize for the confusion this has caused. Under the reviewer’s suggestion, we have now replaced the spectra color scheme with a single-color gradient in Fig. 3b to make it seem better. Additionally, we have incorporated the scale bar by adjusting the positions of images in Fig. 4b.

We greatly appreciate the reviewer’s time and helpful suggestions, which have significantly contributed to improving the quality of our paper. All the changes as be found as below:

“... ”

Fig. 2 ... f, Height distributions and line profiles for non-treated (pink) and SAW-treated (blue) Cu foils extracted from AFM maps in (d) and (e). ...”

* Revised Manuscript P14 Figure 2.

“... ”

Fig. 3 ... b, Raman spectra collected from 400 positions (20 × 20 array) on the graphene film shown in (a). ...”

* Revised Manuscript P15 Figure 3.

“ ...

Fig. 4 ... b-c, Photographs and optical images of hBN monolayers on rough (top) and flat (bottom) Cu foils after the oxidation experiment. ...”

*** Revised Manuscript P16 Figure 4.**